# Exploring the Volatile Composition and Antibacterial Activity of Edible Flower Hydrosols with Insights into Their Spontaneous Emissions and Essential Oil Chemistry

**DOI:** 10.3390/plants13081145

**Published:** 2024-04-19

**Authors:** Basma Najar, Ylenia Pieracci, Filippo Fratini, Laura Pistelli, Barbara Turchi, Dario Varriale, Luisa Pistelli, Maria Francesca Bozzini, Ilaria Marchioni

**Affiliations:** 1RD3—Pharmacognosy, Bioanalysis & Drug Discovery Unit, Analytical Platform of the Faculty of Pharmacy, Faculty of Pharmacy, Free University of Brussels, Blvd Triomphe, Campus Plaine, CP 205/5, B-1050 Brussels, Belgium; 2Department of Pharmacy, University of Pisa, Via Bonanno 6, 56126 Pisa, Italy; yleniapieracci@gmail.com (Y.P.); luisa.pistelli@unipi.it (L.P.); mariafrancesca.bozzini@gmail.com (M.F.B.); 3Department of Veterinary Sciences, University of Pisa, Viale dellle Piagge 2, 56124 Pisa, Italy; filippo.fratini@unipi.it (F.F.); barbara.turchi@unipi.it (B.T.); d.varriale@studenti.unipi.it (D.V.); 4Centro Interdipartimentale di Ricerca Nutraceutica e Alimentazione per la Salute (NUTRA-FOOD), Università di Pisa, Via del Borgetto 80, 56124 Pisa, Italy; laura.pistelli@unipi.it; 5Dipartimento Scienze Agrarie, Alimentari e Agro-ambientali (DISAAA-a), Università di Pisa, Via del Borgetto 80, 56124 Pisa, Italy; 6Department of Food and Drug, University of Parma, Parco Area delle Scienze 27/A, 43124 Parma, Italy; ilaria.marchioni@unipr.it

**Keywords:** *Antirrhinum majus*, *Begonia cucullata*, *Calandula officinalis*, *Dahlia hortensis*, *Polianthes tuberosa*, *Tulbaghia cominsii*

## Abstract

In the circular economy framework, hydrosols, by-products of the essential oil industry, are gaining attention for their potential in waste reduction and resource reuse. This study analyzed hydrosols from six edible flowers, investigating their chemical composition (VOC-Hyd) and antibacterial properties alongside volatile organic compounds of fresh flowers (VOC-Fs) and essential oils (EOs). *Antirrhinum majus* exhibited ketones as major VOC-Fs (62.6%) and VOC-Hyd (41.4%), while apocarotenoids dominated its EOs (68.0%). *Begonia cucullata* showed alkanes (33.7%) and aldehydes (25.7%) as primary VOC-Fs, while alkanes were prevalent in both extracts (65.6% and 91.7% in VOC-Hyd and in EOs, respectively). *Calandula officinalis* had monoterpenoids in VOC-Fs and VOC-Hyd (89.3% and 49.7%, respectively), while its EOs were rich in sesquiterpenoids (59.7%). *Dahlia hortensis* displayed monoterpenoid richness in both VOC-Fs and extracts. Monocots species’ VOC-Fs (*Polianthes tuberosa*, *Tulbaghia cominsii*) were esters-rich, replaced by monoterpenoids in VOC-Hyd. *P. tuberosa* EO maintained ester richness, while *T. cominsii* EOs contained a significant percentage of sulfur compounds (38.1%). Antibacterial assays indicated comparable minimum inhibitory concentration profiles across VOC-Hyd: *B. calcullata* and *P. tuberosa* against *Staphylococcus aureus* and *Salmonella enterica* ser. *typhimurium*, *T. cominsii* against *Escherichia coli* and *S. enterica*, *A. majus* and *C. officinalis* against *S. aureus*, and *D. hortensis* against *S. enterica*.

## 1. Introduction

The deep bond between humans and flowers has been reported for millennia. Historically, the latter were protagonists of funeral rituals, feasts, offerings, and songs due to their myriad symbolic meanings. Their undeniable aesthetic beauty has made them highly sought-after for decorative functions, paintings, ceramics, and fabrics [1]. Gardens, in particular, result from skillful combinations of plant species with a multitude of shapes and colors, whose wonderfulness can be fully observed during blooming. Very recently, the concept of a ‘healing garden’ has emerged, specifically designed to promote well-being, comfort, and pain tolerance both in clinical and non-clinical populations, thereby representing a combination of flower beauty and human health [2].

The flowers’ aroma is pivotal, as much as their ornamental value, since it can evoke several emotional responses tied to individual experiences and memories. This is attributed to the varied bouquets of volatile organic compounds (VOCs) emitted by petals, each contributing to the distinctive signature of every flower. Flower extracts as essential oils (EOs) hold particular appeal for industries such as cosmetics, perfume, toiletries, and hygiene [3].

In recent years, the gastronomic and nutritional value of flowers has garnered attention, with several edible flowers (EFs) recognized for their attractiveness, sensory perception [4,5,6,7], nutritional properties [8], and biological activities [9]. EFs are very versatile and suitable for various culinary contexts, such as raw flowers without processing, by minimum processing, or in powdered form [10]. Furthermore, their EOs are gaining more and more interest from the food industry: (1) detailed scientific reports highlighting the antimicrobial, antioxidant, and antiparasitic activities of EFs EOs are increasing; (2) they could be used as safer substitutes for chemical preservatives, thus meeting consumer concerns about the use of artificial and harmful compounds in the food sector [11]. Therefore, the EOs obtained from EFs are the perfect combination for the preservation of food products and food safety.

Hydrodistillation is widely used to extract EOs from flowers [12]. During this process, three main by-products are created: solid residue (plant debris), water residue (non-distilled aqueous phase), and hydrosols (also known as hydrosols or aromatic water). The latter are aqueous aromatic solutions saturated with water-soluble volatile compounds [13]. Hydrosols distillate together with the EOs, retaining a good quantity of dissolved compounds that are, however, in lesser amounts than EOs, both in terms of number and concentration. Despite this, oxygenated components (e.g., oxygenated monoterpenes (OM)) are higher in hydrosols, while EOs are richer in highly hydrophobic compounds such as monoterpene and sesquiterpene hydrocarbons [13]. Currently, aromatic waters have several applications in the food sector, as well as in the aromatherapy, cosmetics, and perfume industries, representing the most commonly used by-product of EO production [14].

Within this framework, the current study meticulously examined the chemical composition of hydrosols derived from six distinctive flowers tailored for culinary purposes, namely, *Antirrhinum majus* L., *Begonia cucullata* Willd, *Calendula officinalis* L., *Dahlia hortensis* Guillaumin, *Polianthes tuberosa* L., and *Tulbaghia cominsii* Vosa. Our investigation extended beyond the hydrosols, encompassing a thorough analysis of the EOs and spontaneous emissions of these botanical specimens. Furthermore, a comprehensive assessment of the antibacterial properties was conducted, involving six bacterial strains. This study scrutinized the primary phytochemicals inherent in these floral extracts, marking a pivotal stride in revealing the transformative potential encapsulated within these often-neglected by-products.

## 2. Material and Methods

### 2.1. Plant Material

Plants of *Antirrhinum majus* L., *Begonia cucullata* Willd, *Calendula officinalis* L., *Dahlia hortensis* Guillaumin, *Polianthes tuberosa* L., and *Tulbaghia cominsii* Vosa were kindly provided by CREA Research Centre for Vegetable and Ornamental Crops (CREA, Sanremo, Impe-ria, Italy, GPS: 43.816887, 7.758900) and Chambre d’Agriculture des Alpes-Maritimes (CREAM, Nice, France, GPS: 43.668318 N, 7.204194 E). Detailed information on plant cultivation (propagation, substrate composition, fertilization treatment, and frequency and type of irrigation) has already been published by Drava et al. [15] (*A.majus*, *B. cucullata*, *D. hortensis*—see *D. pinnata*, and *T. cominsii*), and Copetta et al. [16] (*P. tuberosa*). *C. officinalis* is part of the edible flower collection cultivated at CREA, as reported by Marchioni et al. [17].

Full-opened and flawless flowers were collected at the Department of Pharmacy (University of Pisa) and properly analyzed as follows.

### 2.2. Essential Oil (EO) and Hydrosol Extraction

The extraction was conducted using 20 g of fresh flowers through a 2 h hydrodistillation process employing a micro-Clevenger-type apparatus. During hydrodistillation, 1 mL of HPLC-grade *n*-hexane was added to the separator funnel. This precaution was taken due to concerns about the possibility of obtaining an EO yield as low as one drop. The extraction process resulted in obtaining both EO dissolved in hexane and hydrosol, which were then separated based on density difference. The dissolved EO was directly analyzed by GC-MS, while the hydrosol was stored in sealed vials at 4 °C in the dark.

### 2.3. Headspace Solid-Phase Microextraction (HS-SPME) for VOC Analyses

The HS-SPME procedure, as outlined by Najar et al. [18], involved both spontaneous emissions of fresh EFs (VOC-Fs) and hydrosols (VOC-Hyd). Briefly, 1 g of fresh flower or 1 mL of hydrosol was placed into a 50 mL headspace glass vial, sealed with aluminum foil, and stored at room temperature. HS-SPME extractions utilized a DVB/CAR/PDMS SPME fiber exposed to the headspace for 30 min. Post-extraction, the fiber was introduced into the GC-MS injection system at 220 °C for 3 min for thermal desorption of analytes. All analyses were performed in triplicate (n = 3). The SPME fiber underwent thermal conditioning before use and daily 10 min conditioning before the initial extraction to ensure the absence of carryover effects.

### 2.4. Phytochemical Analysis: GC-MS Analysis

Chromatographic analyses were conducted on VOC-Fs, EOs, and VOC-Hyd extracted from edible flowers using an Agilent 7890B gas chromatograph system (Agilent Technologies Inc., Santa Clara, CA, USA) equipped with an Agilent HP-5MS capillary column (30 m × 0.25 mm; Santa Clara, CA, USA). Helium served as carrier gas, with a flow rate of 1 mL/min and a column head pressure of 13 psi. The injector temperature was set at 220 °C, with a split ratio of 1:25. The oven temperature programming ranged from 60 to 240 °C at a rate of 3 °C/min. Full scan MS detection was conducted using an Agilent 5977B single quadrupole inert mass selective detector (Agilent Technologies Inc., Santa Clara, CA, USA) with an electron impact (EI) ion energy of 70 eV, and the mass acquisition range was set to 30–300 *m*/*z*.

Identification of the constituents was based on a comparison of the retention times (Rt) with those of the authentic samples, comparing their calculated Kovats Index (KI) (determined using the van Den Dool and Kratz equation) to the series of a C8–C22 *n*-hydrocarbons. Computer matching was also used against commercial [19] and laboratory-developed mass spectra library built up from pure substances and components of commercial essential oils of known composition and MS literature data [20,21,22,23,24]. Analysis was conducted in triplicate for each sample to ensure reliability.

### 2.5. Preparation and Storage of Bacterial Cultures

Three Gram-positive bacteria, *Staphylococcus aureus* ATCC 6538, *Listeria monocytogenes* ATCC 7644, and *Enterococcus faecalis* V583E, and three Gram-negative bacteria, *Escherichia coli* ATCC 15325, *Pseudomonas aeruginosa* ATCC 27853, and *Salmonella enterica* serovar Typhimurium ATCC 14028, were selected as test organisms. The bacterial cultures were prepared in Tryptone Soy Agar (TSA, pH 7.3 ± 0.2, Oxoid, UK) and stored at −20 °C in BHI broth (pH 7.4 ± 0.2, Merck, Germany) with 20% glycerol as cryoconservant. To revitalize bacterial cultures, frozen stocks were subcultured in BHI broth and incubated at 37 °C for 24 h.

#### In Vitro Antibacterial Testing of Hydrosols

MIC values for hydrosols against diverse bacterial strains were determined using a modified twofold serial microdilution method in a 96-well microplate. Hydrosols prepared in BHI with DMSO followed a 1:3:4 *v*/*v* ratio. Incubation at 37 ± 2 °C for 24 h was performed after bacterial suspension addition. MIC values, evaluated in triplicate, were mode-determined for each isolate. MBC values, indicating the lowest hydrosols concentration preventing bacterial growth, were assessed on TSA plates after 24 h incubation. Results, expressed as *v*/*v*, were mode-reported for both MIC and MBC. The entire process was executed in triplicate to ensure reliability. 

### 2.6. Statistical Analysis

Volatile compositions of hydrosols underwent comprehensive multivariate statistical analyses utilizing JMP software (SAS Institute Inc. JMP^®^. Version 16, SAS Institute Inc., Cary, NC, USA, 2021). Principal Component Analysis (PCA) was performed on a covariance data matrix of 58 × 6 (58 compounds × 6 samples = 348 data). The resulting PCA was plotted by selecting the two principal components with the highest variance explained, obtained by the linear regressions operated on mean-centered, unscaled data. As an unsupervised method, this analysis aimed at reducing the dimensionality of the multivariate data of the matrix whilst preserving most of the variance [25]. Additionally, a two-way Hierarchical Cluster Analysis (HCA) was carried out using Ward’s method, and the squared Euclidean distances were used as a measure of similarity.

## 3. Results and Discussion

### 3.1. Phytochemical Insight into the Studied Species

#### 3.1.1. *Antirrhinum majus* L.

Spontaneous emission of its flowers was characterized by ketones (62.6%), mainly acetophenone (59.9%, (15)) (Table 1), reported for its sweet, pungent odor and taste [26]. This class of constituents was followed by esters (10.8%) and OM (9.2%), of which methyl benzoate (5.4%, (17)) and linalool (8.3%, (18)) were, respectively, the most abundant compounds. In coherence with our results, Suchet et al. [27] reported the presence of acetophenone in the floral scent of wild snapdragon, with a contribution of 69.0% of the absolute emissions.

EO evidenced a lesser number of constituents (13) in comparison with spontaneous emission (18) and VOC-Hyd (21). Sixty-eight percent of the identified portion was represented by apocarotenoids uniquely constituted by hexahydrofarnesylacetone (40). A significant decrease in acetophenone was noticed after hydrodistillation, whose percentage passed from 59.3% in VOC-Fs to 5.7% in EO.

Ketone turned out to be the main class in VOC-Hyd (41.4%), and acetophenone (40.2%) was also confirmed as the main compound by excellence here. Aldehydes constitute more than a quarter of the whole VOC-Hyd composition (26.7%), with nonanal being the main constituent (23.6%, (19)).

#### 3.1.2. *Begonia cucullata* Willd

The spontaneous emission of begonia pointed out only six compounds, including one alkane (tetradecane (33.7%, (16)), one aldehyde (decanal (25.7%, (14)), and one ester (benzyl acetate (7.6%, (12)), along with three terpenoids, collectively accounting for over 30% of the total identified fraction (Table 2).

The EO yielded eight identified compounds, with five belonging to the alkane class, making up the predominant category at 91.7%.

VOC-Hyd, however, exhibited the highest diversity, featuring 15 identified constituents. Aldehydes prevailed in the composition (65.6%), with nonanal being the chief compound (56.9%, (10)). Notably, both nitrogenous compounds (NCs, 9.8%) and MH (9.9%) were equally represented, with oxime and methoxy phenyl (2) exclusively representing the NCs. Limonene (7.5%, (6)) emerged as the principal MH. As not much research has been performed so far on *B. cucullata*, making direct comparisons is challenging with only *B. reniformis* Dryand. Leaf EO was reported in a study by Da Silva et al. [28]. Sesquiterpenoids siliphiperfol-4,7(14)-dine and β-vetispirene were major constituents, constituting 15.7 and 21.0%, respectively. The unique OS compound in the current work was viridiflorol, albeit in a minimal percentage (0.7%, (20)).

#### 3.1.3. *Calendula officinalis* L.

Spontaneous emissions of *C. officinalis* were rich in monoterpene hydrocarbons (MH) (49.5%), primarily represented by α-thujene (44.8%, (3)) (Table 3). Additionally, the presence of sesquiterpene hydrocarbons (39.6%) was noted, predominantly represented by δ-cadinene (43) and γ-cadinene (41), comprising 15.3% and 11.1%, respectively (Table 3). α-Thujene was also the main compound found in the SEs of *C. arvensis* L. [5], which also highlighted the presence of considerable amounts of sesquiterpenes in this species.

The EO contained sesquiterpenes, especially oxygenated ones (42.7% vs. 17.0% hydrocarbons). The main constituents were α-cadinol (18.8%, (49)) and *tau*-cadinol (16.1%, (47)). It is interesting to note the presence of non-terpenes in significant amounts, representing 29.2% of the identified fraction. α-Cadinol emerged as the predominant constituent identified in Bosnians *C. officinalis* flowers studied by Ak et al. [29]. Additionally, it is noted as one of the principal compounds in this species, as reported by Dhingara et al. [30].

The by-product of EO extraction was rich in OM (79.9%), primarily represented by eucalyptol (41.4%, (9)) and linalool (12.2%, (14)). However, previous studies on *C. arvensis* VOC-Hyd have reported a prevalence of oxygenated compounds [31].

#### 3.1.4. *Dahlia hortensis* Guillaumin

The spontaneous emission of *D*. *hortensis* flowers was dominated by terpene compounds, especially monoterpene and sesquiterpene hydrocarbons (61.1% and 38.6%, respectively). Monoterpene hydrocarbons were mainly represented by *p*-cymene (46.6% (11)) and α-phellandrene (12.1%, (9)) (Table 4). Meanwhile, sesquiterpene hydrocarbons were mainly represented by germacrene D (14.1%, (24)) and β-caryophyllene (11.3%, (19)) (Table 4). This trend was also observed in the EO, with the main classes being the same as those observed in the VOC-Fs, with a slight decrease in their amounts (54.6% and 21.8%, respectively, in MH and SH). It is interesting to note the presence of alkanes in this extract, which represented 15.2% of the identified fraction, mainly *n*-pentacosane (8.6%, (41)).

Regarding the major compounds found in the EO, we highlighted the presence of *p*-cymene (18.9%, (11)) and limonene (19.3%, (12)). The EO also showed the largest number of compounds (34) compared to both the molecules that were spontaneously perceived (12) and the VOC-Hyd (13). The latter was dominated by MH (95.5%), and *p*-cymene (71.3%) was again confirmed to be the molecule par excellence of these flowers, regardless of the type of extract. Besides *p*-cymene, limonene was also found to have a high percentage of VOC-Hyd (19.2%).

To the best of the authors’ knowledge, no previous paper has been published about the volatilome of this plant. The available literature primarily investigates other species within the same genus, with a predominant focus on EOs. A unique paper investigated *D. pinnata* Cav. specifically for its VOC-Fs composition using static headspace volatiles extraction and revealed the extract’s richness in myrcene (28.5%), γ-muurolene (27.8%), and (*E*)-β-ocimene (17.5%) [32]. The method used cannot be directly compared with our extraction technique or our approach to spontaneous emission evaluation. Nevertheless, it is noteworthy that the primary compounds belonged to MH and SH, as observed herein in both VOC-Fs and EO. Flower EO of *D. pinnata* was also investigated by Wang et al. [33], who reported an EO rich in 4-terpineol (25.7%), methallyl cyanide (14.0%), and D-limonene (10.5%). Only limonene was found in our EO, while the other two compounds were omitted. Within the same genus, the capitulum (flower head) EO of *Dahlia imperialis* Roezel ex Ortgies was rich in β-pinene (27.7%), α-phellandrene (26.2%), and α-pinene (12.4%). A recent study on flower EO of the same species confirmed its dominance of β-pinene (27.7%), α-phellandrene (26.2%), and α-pinene (12.4%) as major chemicals [34]. Although all these compounds were also present in our studied flowers, they were found in lesser amounts. In a study conducted by Manah et al. [35] on the flowers EO of *Dahlia* E‘veline’, it was evidenced that more than 80% of the identified fraction was composed of anethole. However, this compound was omitted in the studied species.

**Table 4 plants-13-01145-t004:** Analysis of spontaneous emissions of fresh flowers (VOC-Fs), essential oils (EO), and hydrosols (VOC-Hyd) derived from *Dahlia hortensis*.

No.	Compounds	Formula	Class	LRI ^cal^	LRI ^lit^	VOC-Fs	EOs	VOC-Hyd
						Relative Abundance (%)		
1	hexanal	C_6_H_12_O	ADH	802	8008 ^1^	-	-	2.2 ± 0.15
2	methoxy-phenyl-oxime	C_8_H_9_NO_2_	NC	898	899 *	-	-	0.2 ± 0.08
3	heptanal	C_7_H_14_O	ADH	901	904 ^1^	-	-	0.8 ± 0.27
4	α-thujene	C_10_H_16_	MH	933	931 ^1^	-	-	0.2 ± 0.07
5	α-pinene	C_10_H_16_	MH	941	937 ^1^	-	0.8 ± 0.16	-
6	sabinene	C_10_H_16_	MH	977	976 ^1^	-	2.2 ± 0.43	1.2 ± 0.08
7	β-pinene	C_10_H_16_	MH	982	980 ^1^	-	2.0 ± 0.36	2.1 ± 0.16
8	β-myrcene	C_10_H_16_	MH	991	990 ^1^	-	0.5 ± 0.11	-
9	α-phellandrene	C_10_H_16_	MH	1006	1007 ^1^	12.1 ± 0.66	5.6 ± 0.50	-
10	α-terpinene	C_10_H_16_	MH	1020	1016 ^1^	-	-	0.2 ± 0.04
11	*p*-cymene	C_10_H_14_	MH	1028	1026 ^1^	46.6 ± 2.03	18.9 ± 0.86	71.3 ± 0.21
12	limonene	C_10_H_16_	MH	1029	1033 ^1^	-	16.3 ± 1.30	19.2 ± 0.67
13	(*E*)-β-ocimene	C_10_H_16_	MH	1052	1050 ^1^	2.4 ± 0.15	8.2 ± 1.50	1.0 ± 0.04
14	γ-terpinene	C_10_H_16_	MH	1058	1053 ^1^	-	-	0.3 ± 0.08
15	cosmene	C_10_H_14_	MH	1131	1134 ^1^	-	0.1 ± 0.04	-
16	4-terpineol	C_10_H_18_O	OM	1177	1171 ^1^	-	-	0.2 ± 0.04
17	thymol methyl ether	C_11_H_16_O	OM	1235	1234 ^1^	-	0.1 ± 0.01	0.3 ± 0.06
18	α-copaene	C_15_H_24_	SH	1376	1372 ^1^	3.5 ± 0.29	0.5 ± 0.02	-
19	β-caryophyllene	C_15_H_24_	SH	1419	1418 ^1^	11.3 ± 0.49	5.8 ± 0.14	-
20	α-humulene	C_15_H_24_	SH	1455	1455 ^1^	0.7 ± 0.08	0.9 ± 0.03	-
21	(*E*)-β-farnesene	C_15_H_24_	SH	1458	1454 ^1^	-	0.6 ± 0.02	-
22	*cis*-muurola-4(14),5-diene	C_15_H_24_	SH	1463	1468 ^1^	0.5 ± 0.04	-	-
23	γ-muurolene	C_15_H_24_	SH	1477	1477 ^1^	0.90.07	0.8 ± 0.04	-
24	germacrene D	C_15_H_24_	SH	1481	1480 ^1^	14.1 ± 1.07	8.6 ± 0.64	-
25	*epi*-cubebol	C_15_H_24_O	OS	1493	1494 ^1^	-	0.5 ± 0.04	-
26	bicyclo-germacrene	C_15_H_24_	SH	1496	1494 *	1.8 ± 0.40	0.9 ± 0.06	-
27	α-muurolene	C_15_H_24_	SH	1499	1499 ^1^	0.4 ± 0.03	0.5 ± 0.03	-
28	7-*epi*-α-selinene	C_15_H_24_	SH	1517	1514 ^1^	-	0.5 ± 0.05	-
29	δ-cadinene	C_15_H_24_	SH	1524	1524 ^1^	5.4 ± 0.61	2.7 ± 0.23	-
30	germacrene D-4-ol	C_15_H_26_O	OS	1575	1578 ^1^	-	0.8 ± 0.09	-
31	caryophyllene oxide	C_15_H_24_O	OS	1581	1582 ^1^	-	0.5 ± 0.07	-
32	copaborneol	C_15_H_26_O	OS	1600	1597 ^3^	-	1.1 ± 0.17	-
33	1-*epi*-cubenol	C_15_H_26_O	OS	1627	1623 ^1^	-	0.7 ± 0.11	-
34	caryophylla-4(14),8(15)-dien-5-ol	C_15_H_24_O	OS	1637	1631 ^1^	-	0.3 ± 0.06	-
35	*tau*-cadinol	C_15_H_26_O	OS	1641	1638 ^1^	-	0.1 ± 0.00	-
36	ylangenol	C_15_H_24_O	OS	1667	1666 ^1^	-	0.4 ± 0.07	-
37	*ent*-germacra-4(15),5,10(14)-trien-1-β-ol	C_15_H_24_O	OS	1695	1694 ^1^	-	2.0 ± 0.44	-
38	xanthorrhizol	C_15_H_22_O	OS	1753	1754 ^1^	-	0.3 ± 0.09	-
39	tricosane	C_23_H_48_	ALK	2300	2300 ^1^	-	4.4 ± 0.33	-
40	*n*-tetracosane (c24)	C_24_H_50_	ALK	2400	2400 ^1^	-	2.2 ± 0.44	-
41	*n*-pentacosane (c25)	C_25_H_52_	ALK	2500	2500 ^1^	-	8.6 ± 0.89	-
	Number of Identified Compounds					12	34	13
	Class of Compounds					VOC-Fs	EOs	VOC-Hyd
	Monoterpene Hydrocarbons (MHs)					61.1 ± 0.95	54.6 ± 0.82	95.5 ± 0.22
	Oxygenated Monoterpenes (OMs)					-	0.1 ± 0.01	0.5 ± 0.18
	Sesquiterpene Hydrocarbons (SHs)					38.6 ± 0.65	21.8 ± 0.18	-
	Oxygenated Sesquiterpenes (OSs)					-	6.7 ± 0.10	-
	Aldehydes (ADHs)					-	-	3.0 ± 0.22
	Alkanes (ALKs)					-	15.2 ± 0.60	-
	Nitrogenous Compounds (NCs)					-	-	0.2 ± 0.08
	Non-terpenes					0.70 ± 0.800	15.2 ± 0.60	3.2 ± 0.14
	Total Identified					99.7 ± 0.80	98.4 ± 0.34	99.2 ± 0.19

LRI ^cal^: Linear Retention Index calculated LRI ^lit^; Linear Retention Index reported in the literature; ^1^: NIST 2014 (National Institute of Standards Technology (www.nist.gov) visited 24 February 2024); ^3^: El-Din et al., 2022 [36]; * Pherobase.com.

#### 3.1.5. *Polianthes tuberosa* L.

*P. tuberosa* spontaneous emissions predominantly consisted of esters (76.5%), mirroring its EO composition (90.1%). Methyl benzoate (57.3%, (14)) was the main ester in spontaneous emissions, constituting, together with methyl salicylate (13.0%, (18)), over 70% of the identified portion (Table 5). The presence of lactones (14.4%) was observed, reported uniquely by two compounds: jasminelactone (13.8%, (27)) and δ-decalactone (0.6%, (26)) (Table 5). It is important to highlight that this study partially differs from previous research, which reported the presence of methyl benzoate and methyl salicylate in six out of seven studied cultivars of *P. tuberosa* [37,38]. Methyl salicylate was present in the studied species. Kutty and Mitra [38] reported the presence of lactones. Even though the identified lactones were different from the ones reported in this work, their presence is similar to what was reported herein.

On the contrary, the main esters found in the EO were methyl icosanoate (24.4%, (3*2*)) and methyl heneicosanoate (58.4%, (33)). Additionally, chromene compounds (precocene II (28)) were detected, contributing only 4.0% to the overall composition. The EO composition in half-opened and fully-opened flowers was rich in methyl benzoate (37.9 and 28.6%, respectively) [39], a compound also present in our flowers but in a lesser amount (7.3%, (13)).

Analyzing the VOC-Hyd composition revealed that 40.0% of the compounds belonged to the OM class, with eucalyptol being the predominant compound at 38.1% (10). NCs accounted for 22.9%, with oxime, methoxy-phenyl (15.5%, (3)), and 2,4,5-trimethyl oxazole (7.4%, (1)) being the only identified compounds. The high percentage of MH (16.8%) is mainly represented by *p*-cymene (8.9%, (8)) and limonene (5.7%, (9)).

#### 3.1.6. *Tulbaghia cominsii* Vosa

Esters (37.2%) and ketones (30.3%) predominate in the chemical composition of *T. cominsii*’s VOC-Fs. The chief ketone compound was acetoveratrone (28.4%, (45)), while benzyl benzoate (14.5%, (50)) and benzyl acetate (10.6%, (29)) were the principal esters (Table 6). *T. simmleri* Beauverd VOC-Fs, as investigated by Marchioni et al. [40], showed a different profile, characterized by spontaneous emissions rich in OM (63.8%), primarily represented by eucalyptol (53.1%) and linalool (15.5%), compounds completely absent in the studied species.

Sulfur compounds took precedence as the main class in the EO, constituting 38.1%, with disulfide, methyl (methylothio) methyl (25.8%, (25)) as the major constituent. Additionally, alkanes (22.9%) and OM (18.3%) exhibited high relative abundance represented by *n*-heneicosane (22.9%, (57)) and thymol (16.3%, (39)).

VOC-Hyd demonstrated a distinct chemical composition, emphasizing a significant amount of monoterpene compounds (46.9% and 22.1% in OM and MH, respectively). Key compounds of these monoterpene constituents include eucalyptol (21.4%, (13)), thymol (19.1%, (39)), and limonene (11.4%, (11)). Furthermore, aldehydes contribute (16.4%) to the overall composition, mostly represented by decanal (9.6%, (34)) and nonanal (5.8%, (22)).

The presence of sulfur compounds was previously reported and is responsible for the characteristic alliaceous smell and taste of *Tulbaghia* species, of which *Tulbaghia violacea* Harv. is probably the most studied one so far [41]. Its EO confirmed the presence of sulfur compounds, mainly represented by 2,3,5-trithiahexane and 2,4,5,7-tetrathiaoctane [42]. This is aligned partially with the current result, where only the latter compound was identified with a non-negligible percentage in the EO (10.0%, (44)). The same study also reported the presence of limonene, eucalyptol, and 4-terpineol in both EO and hexane extracts. However, these compounds were only found in the hydrosol of the studied species. (See Table 6).

### 3.2. Antibacterial Activity of Hydrosols

Due to the limited availability of plant material and consequently very low yields of EOs, the hydrosols, a by-product obtained during the removal of volatile oil through steam distillation [43], underwent antibacterial activity testing. The tests were conducted on six strains, three of which were Gram-positive and three were Gram-negatives.

Among the Gram-positive strains, *Staphylococcus aureus* exhibited significant susceptibility, displaying a MIC mode value of 1:2 (Table 7) for all tested hydrosols, except for *A. majus* and *D. hortensis*. Nonanal and decanal, two aldehydes from green leave volatiles family, were observed in all active hydrosols, except for *T. cominsii*, where nonanal was completely omitted; these phytochemicals have been granted a ‘generally recognized as safe’ status [44]. The antibacterial efficacy of decanal was assessed against *S. aureus* strains, including both methicillin-resistant and susceptible strains. However, its effect was less pronounced when compared to the EO of *Ducrosia anethifolia* Boiss, where decanal was the primary compound [45]. Nonanal, a saturated aldehyde, has been documented to induce notable changes in membrane permeability, functioning as an effective antibacterial agent [46].

Limonene, present in all studied species, exhibited significant inhibition of *S. aureus* growth [47] and demonstrated activity against isolated methicillin-resistant strains [48]. Additionally, *p*-cymene, a precursor of carvacrol, present in *C. officinalis* hydrosols, and γ-terpinene, identified in all active hydrosols (AH), displayed antibacterial and anti-biofilm activities [49]. Furthermore, eucalyptol, found in substantial amounts in the hydrosols of *P. tuberosa* and *C. officinalis*, has reported effects on membrane integrity and implications in oxidative stress in methicillin-resistant *S. aureus* [50].

As regards Gram-negative bacteria, *Salmonella enterica* emerged as the most sensitive strain, showing susceptibility to the hydrosols of *B. cucullata*, *D. hortensis*, *P. tuberosa*, and *T. cominsii*. Additionally, the hydrosol of *T. cominsii* exhibited a MIC value of 1:2 against *E. coli*. Fenchone, the primary compound detected in *T. cominsii*, was the focal point of a study evaluating its antibacterial and anti-biofilm properties through in vitro and in silico approaches. In silico predictions revealed interactions with *E. coli* proteins, which were validated by determining MIC and MBC values [51]. The same work evidenced that fenchone reduced biofilm formation in *E. coli*. Additionally, the MIC value of this phytochemical against *E. coli* was 2 [52]. On the other hand, the same hydrosol was rich in viridiflorol, known for its anticancer, antioxidant, anti-inflammatory, and antibacterial activities [52,53,54].

Methoxyphenyl-oxime, found in a high percentage in *P. tuberosa* but also present in all other hydrosols except for *D. hortensis*, albeit in lesser abundance, is an alkaloid reported to be isolated from *Conocarpus lancifolius* Engl. [55]. The authors of this paper demonstrated its antibacterial activity against Gram-negative strains. Further research [56] explored its antiviral effects and confirmed its potential as a potent drug-like compound against capripox viruses, utilizing methanolic extract of *Leucas aspera*.

### 3.3. The Multivariate Analysis of Hydrosol and Key Compound Insights

In addition to assessing their antibacterial activity, hydrosols are being explored as valuable by-products, requiring a comprehensive evaluation of their composition. This necessitates the use of advanced analytical tools such as multivariate analysis, including Hierarchical Cluster Analysis (HCA) and Principal Component Analysis (PCA), which provide profound insights into the characteristics of hydrosols. HCA (Figure 1a) analysis revealed two main clusters, with the first one being homogenous, comprising uniquely *D. hortensis* hydrosol. The second group encompassed the remaining samples, which can be further divided into two subclusters: Subcluster 1, including *A. majus* and *B. cucullata*, and the second subcluster comprised the others. According to the PCA plot (Figure 1b), the first axis (PC 1), accounting for over 45% of the variability, clearly distinguished *D*. *hortensis* hydrosol from the others.

Meanwhile, PC 2 (accounting for 34.5% of the variability) differentiated *A. majus* and *B. cucullata* from the rest. These findings were corroborated by the cluster analysis, where the Ward method clustered *D. hortensis* separately, while *A. majus* and *B. cucullate* formed another cluster, and the remaining species grouped, including *T. cominsii*, *P. tuberosa*, and *C. officinalis*.

Examining the biplot of the PCA analysis (Figure 1c), we observed that *D*. *hortensis* stood out due to its high percentage in *p*-cymene, a common compound found in all aromatic waters. *p*-cymene, an alkyl-substituted aromatic compound, is known for its diverse beneficial activities, such as antioxidant, anti-inflammatory, antiparasitic, antidiabetic, antiviral, antitumor, antibacterial, and antifungal activities [57]. *p*-Cymene has also been recognized to act as an analgesic, antinociceptive, immunomodulatory, vasorelaxant, neuroprotective agent, and anticancer [57]. Moreover, it found applications in the food industry as a flavor/fragrance agent [58] and served as an intermediate constituent in the chemical syntheses of fragrances [59]. Additionally, sabinene and (*E*)-β-ocimene were exclusively present in *Dahlia* aromatic water. These compounds represented 1% of the identified fraction. (*E*)-β-ocimene was recommended for fragrance use at levels up to 3% and is known as a pheromone involved in social regulation in the honeybee colony [60]. Sabinene was utilized as a perfume additive and possessed anti-fungal activity [61]. Also noticed is the presence of β-pinene reported for broad biological activity, including fungicidal, antimicrobial, and antiviral agents, in addition to its use in the flavor and fragrance industry [62].

Nonanal, a common compound found in both *A. majus* and *B. cucullata*, contributed to their clustering in the bottom left quadrant of the PCA analysis. It is renowned for its antifungal properties [63] and is widely used in perfumery products for its green-floral fragrance [64]. Acetophenone, a preliminary compound of *A. majus*, serves as both an attractant and repellent for blood-sucking insects such as mosquitoes, flies, and ticks, serving as a crucial component in the manipulation of skin microbiota by vector-borne parasites and as an ingredient in trap crops for economically important crop pests [26]. Furthermore, it serves as a flavoring agent and intermediate compound in perfumes and cosmetics [65].

On the other hand, 1-octene-3-ol and bornyl acetate were characteristic compounds of *B. cucullata*. They exhibited strong antibacterial activity, especially against Gram-positive strains [66,67], which could be one of the reasons why the hydrosols of this species were effective on *S. aureus*. Bornyl acetate is also documented for its insecticidal activity [68] and anti-inflammatory effect in human chondrocytes [69]. On the contrary, *A. majus* was characterized by 1-hexanol, a volatile alcohol known to have an effect against food-related Gram-negative bacteria [70].

In the upper left quadrant of PCA analysis, the remaining hydrosols were clustered together due to their high content in eucalyptol, which is known for its pleasant spicy aroma and taste. Eucalyptol found applications as a flavoring agent, fragrance ingredient, and cosmetics additive [71]. Additionally, it is reported to alleviate pain and inflammation associated with monosodium urate [72]. Notably, methoxy-phenyl-oxime constituted at least 5.0% of the identified fraction of these three hydrosols and was found in all the others, imparting a scent reminiscent of fresh shrimp and crabs [73]. This compound also exhibited antibacterial activity [55]. Oximes naturally occur in plants and animals and are known for their anti-inflammatory, antioxidant, and anticancer activities [74]. Thymol, a characteristic constituent of *T. cominsii*, is versatile and used in therapeutic applications [75] and food [76]. Decanal, another prominent compound in this hydrosol, is utilized in perfume [77]. In contrast, methyl benzoate and 6-methyl-5-hepten-2-one were present in *P. tuberosa*. Methyl benzoate is suggested as an efficient green pesticide [78]. 6-Methyl-5-hepten-2-one is considered an important chemical intermediate and flavor compound crucial for fruit flavor in tomato, papaya, and guava [79]. *C. officinalis* evidenced the highest content of linalool, a compound mainly used for its anti-inflammatory, anticancer, antimicrobial, and antioxidant properties, alongside its use in cosmetics, food additives, and perfume [77,80]. Furthermore, this hydrosol is characterized by its high content of camphor and 4-terpineol, both of which are reported for their antibacterial activity [81,82].

Limonene, a shared compound in all hydrosols, although it may not be considered a discriminant compound in PCA due to its presence in significant percentages across all samples, is recognized for its therapeutic effect, including anti-inflammatory, antioxidant, and antiviral effects, besides its use as flavor and fragrance additive owing to its pleasant lemon-like odor [83].

Overall, the studied hydrosols offer various chemical compositions with promising applications across multiple industries, including skincare, fragrance, and therapeutics.

## Figures and Tables

**Figure 1 plants-13-01145-f001:**
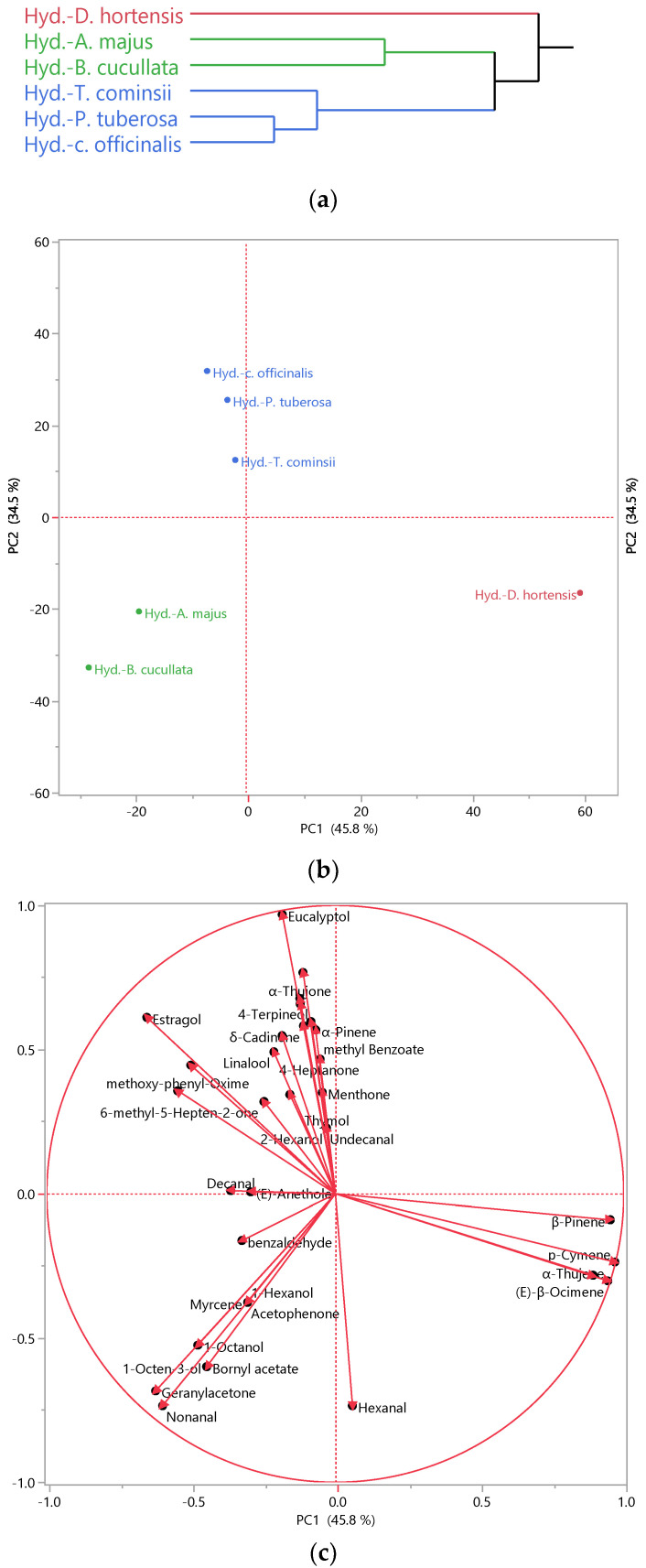
Hierarchical Cluster Analysis (HCA) (**a**), Principal Component Analysis (PCA) performed on the hydrosols of the studied species. (**b**): Score plot; (**c**): loading plot.

**Table 1 plants-13-01145-t001:** Analysis of spontaneous emissions of fresh flowers (VOC-Fs), essential oils (EO), and hydrosols (VOC-Hyd) derived from *Antirrhinum majus*.

No.	Compounds	Formula	Class	LRI ^cal^	LRI ^lit^	VOC-Fs	EOs	VOC-Hyd
						Relative Abundance (%)		
1	2,4,5-trimethyl oxazole	C_6_H_9_NO	NC	852	850 ^1^	-	-	2.5 ± 0.12
2	1-hexanol	C_6_H_14_O	ALC	871	873 ^1^	-	-	4.5 ± 0.17
3	methoxy-phenyl-oxime	C_8_H_9_NO_2_	NC	898	899 *	-	-	4.0 ± 0.22
4	benzaldehyde	C_7_H_6_O	ADH	962	969 ^1^	-	-	1.1 ± 0.09
5	1-octen-3-one	C_8_H_14_O	KET	975	978 ^1^	-	1.0 ± 0.08	-
6	β-pinene	C_10_H_16_	MH	982	980 ^1^	-	-	0.2 ± 0.06
7	6-methyl-5-hepten-2-one,	C_8_H_14_O	KET	986	987 ^1^	-	-	1.0 ± 0.04
8	myrcene	C_10_H_16_	MH	991	990 ^1^	-	-	0.2 ± 0.09
9	1-hexyl acetate	C_8_H_16_O	EST	1012	1015 ^1^	-	0.9 ± 0.37	-
10	*p*-cymene	C_10_H_14_	MH	1028	1026 ^1^	-	-	2.8 ± 0.07
11	limonene	C_10_H_16_	MH	1029	1033 ^1^	-	-	5.3 ± 0.11
12	eucalyptol	C_10_H_18_O	OM	1032	1033 ^1^	-	-	2.5 ± 0.07
13	(*E*)-β-ocimene	C_10_H_16_	MH	1052	1050 ^1^	4.3 ± 0.68	-	-
14	γ-terpinene	C_10_H_16_	MH	1058	1053 ^1^	-	-	0.5 ± 0.02
15	acetophenone	C_8_H_8_O	KET	1065	1066 ^1^	59.3 ± 1.00	5.7 ± 0.65	40.2 ± 0.85
16	methyl benzoate	C_8_H_8_O_2_	EST	1092	1091 ^1^	-	0.4 ± 0.07	-
17	methyl ester-benzoic acid	C_8_H_8_O_2_	EST	1094	1091 ^1^	5.4 ± 0.12	-	-
18	linalool	C_10_H_18_O	OM	1101	1094 ^1^	8.3 ± 0.46	2.3 ± 0.39	3.8 ± 0.04
19	nonanal	C_9_H_18_O	ADH	1104	1101 ^1^	-	0.9 ± 0.16	23.6 ± 0.03
20	methyl nicotinate	C_7_H_7_NO_2_	PYR	1139	1137 ^1^	0.4 ± 0.00	-	-
21	camphor	C_10_H_16_O	OM	1142	1143 ^1^	-	-	0.1 ± 0.01
22	1-phenyl-2-Propen-1-one	C_9_H_8_O	KET	1143	1047 ^1^	-	1.8 ± 0.31	-
23	2-hydroxyacetophenone	C_8_H_8_O_2_	KET	1162	1167 ^1^	3.3 ± 0.24	-	-
24	dimethoxybenzene	C_8_H_10_O_2_	ETR	1190	1192 ^1^	2.9 ± 0.32	-	-
25	methyl salicylate	C_8_H_8_O_3_	EST	1192	1190 ^1^	1.8 ± 0.03	-	-
26	estragol	C_10_H_12_O	OM	1196	1997 ^1^	-	-	1.0 ± 0.03
27	decanal	C_10_H_20_O	ADH	1206	1208 ^1^	-	-	2.0 ± 0.01
28	2-methyl-2-nonen-4-one	C_10_H_18_O	KET	1213	1215 ^1^	-	-	0.2 ± 0.13
29	3,5-dimethoxytoluene	C_9_H_12_O_2_	ETR	1274	1276 ^1^	2.9 ± 0.23	0.6 ± 0.02	1.9 ± 0.03
30	(*e*)-anethole	C_10_H_12_O_2_	PP	1286	1284 ^1^	-	0.6 ± 0.00	0.6 ± 0.02
31	thymol	C_10_H_14_O	OM	1291	1292 ^1^	0.9 ± 0.05	1.9 ± 0.14	-
32	geranylacetone	C_13_H_22_O	AC	1456	1457 ^1^	-	-	0.4 ± 0.00
33	*(e,e)*-α-farnesene	C_15_H_24_	SH	1507	1506 ^1^	1.2 ± 0.35	-	-
34	viridiflorol	C_15_H_26_O	OS	1591	1593 ^1^	2.1 ± 0.05	6.6 ± 0.22	-
35	hedione	C_13_H_22_O_3_	EST	1649	1648 ^1^	1.8 ± 0.12	-	-
36	precocene ii	C_13_H_16_O_3_	CHR	1658	1656 ^1^	1.0 ± 0.20	-	-
37	coumarin derivative	C_19_H_18_O_2_	LAC	1658		1.0 ± 0.02	-	-
38	2-hexyl-(*e*)-cinnamaldehyde	C_15_H_20_O	ADH	1749	1754 ^1^	0.2 ± 0.04	-	-
39	*iso*-propyl myristate	C_17_H_34_O_2_	EST	1827	1824 ^1^	1.8 ± 0.17	-	-
40	hexahydrofarnesylacetone	C_18_H_36_O	AC	1844	1847 ^1^	1.4 ± 0.03	68.0 ± 2.70	-
41	phytol	C_20_H_40_O	OD	2114	2119 ^1^	-	4.0 ± 0.29	-
	Number of Identified Compounds					18	13	21
	Class of Compounds					VOC-Fs	EOs	VOC-Hyd
	Monoterpene Hydrocarbons (MHs)					4.3 ± 0.68	-	9.0 ± 0.11
	Oxygenated Monoterpenes (OMs)					9.2 ± 0.25	4.2 ± 0.35	7.4 ± 0.04
	Sesquiterpene Hydrocarbons (SHs)					1.2 ± 0.35	-	-
	Oxygenated Sesquiterpenes (OSs)					2.1 ± 0.05	6.6 ± 0.22	-
	Oxygenated Diterpenes (ODs)					-	4.0 ± 0.29	-
	Apocarotenoids (ACs)					1.4 ± 0.03	68.0 ± 2.70	0.4 ± 0.00
	Phenylpropanoids (PPs)					-	0.6 ± 0.00	0.6 ± 0.02
	Aldehydes (ADHs)					0.2 ± 0.04	0.9 ± 0.16	26.7 ± 0.13
	Alcohols (ALCs)					-	-	4.5 ± 0.17
	Chromene Compounds (CHRs)					1.0 ± 0.20	-	-
	Esters (ESTs)					10.8 ± 0.44	1.3 ± 0.21	-
	Ethers (ETRs)					5.8 ± 0.28	0.6 ± 0.02	1.9 ± 0.03
	Ketones (KETs)					62.6 ± 0.75	8.5 ± 0.65	41.4 ± 0.34
	Lactones Compounds (LACs)					1.0 ± 0.02	-	-
	Nitrogenous Compounds (NCs)					-	-	6.5 ± 0.15
	Pyridines (PYRs)					0.4 ± 0.00	-	-
	Non-terpenes					81.8 ± 0.25	11.3 ± 0.26	81.0 ± 0.17
	Total Identified					100.0 ± 00	94.7 ± 0.51	98.4 ± 0.09

LRI ^cal^: Linear Retention Index calculated LRI ^lit^; Linear Retention Index reported in the literature; ^1^: NIST 2014 (National Institute of Standards Technology (www.nist.gov) visited 24 February 2024); * Pherobase.com.

**Table 2 plants-13-01145-t002:** Analysis of spontaneous emissions of fresh flowers (VOC-Fs), essential oils (EOs), and hydrosols (VOC-Hyds) derived from *Begonia cucullata*.

No.	Compounds	Formula	Class	LRI ^cal^	LRI ^lit^	VOC-Fs	EOs	VOC-Hyd
						Relative Abundance (%)		
1	hexanal	C_6_H_12_O	ADH	802	800 ^1^	-	-	4.1 ± 0.25
2	methoxy-phenyl-oxime	C_8_H_9_NO_2_	NC	898	899 *	-	-	9.8 ± 0.79
3	1-octen-3-ol	C_8_H_16_O	ALC	981	979 ^1^	-	-	3.1 ± 0.08
4	5-hepten-2-one, 6-methyl-	C_8_H_14_O	KET	986	987 ^1^	-	-	1.0 ± 0.14
5	*p*-cymene	C_10_H_14_	MH	1028	1026 ^1^	-	-	1.7 ± 0.02
6	limonene	C_10_H_16_	MH	1029	1033 ^1^	-	-	7.5 ± 0.03
7	eucalyptol	C_10_H_18_O	OM	1032	1033 ^1^	-	-	1.2 ± 0.12
8	γ-terpinene	C_10_H_16_	MH	1058	1053 ^1^	-	-	0.7 ± 0.04
9	1-octanol	C_8_H_18_O	ALC	1071	1074 ^1^	-	-	2.9 ± 0.51
10	nonanal	C_9_H_18_O	ADH	1104	1101 ^1^	-	3.4 ± 0.44	56.9 ± 1.70
11	camphor	C_10_H_16_O	OM	1142	1143 ^1^	-	-	0.9 ± 0.09
12	benzyl acetate	C_9_H_10_O_2_	EST	1164	1162 ^1^	7.6 ± 0.14	-	-
13	estragol	C_10_H_12_O	OM	1196	1997 ^1^	-	-	0.6 ± 0.03
14	decanal	C_10_H_20_O	AD	1206	1208 ^1^	25.7 ± 2.28	-	4.6 ± 0.26
15	bornyl acetate	C_12_H_20_O_2_	OM	1285	1284 ^1^	-	-	1.9 ± 0.07
16	tetradecane	C_14_H_30_	ALK	1400	1400 ^1^	33.7 ± 0.03	-	-
17	β-caryophyllene	C_15_H_24_	SH	1419	1418 ^1^	8.3 ± 0.12	-	-
18	geranylacetone	C_13_H_22_O	AC	1456	1457 ^1^	-	-	0.8 ± 0.04
19	5,9-Undecadien-2-one, 6,10-dimethyl- (*trans*-geranylacetone)	C_13_H_22_O	KET	1456	1453 ^1^	13.7 ± 0.52	-	-
20	viridiflorol	C_15_H_26_O	OS	1591	1593 ^1^	8.8 ± 0.10	0.7 ± 0.19	-
21	precocene ii	C_13_H_16_O_3_	CHR	1658	1656 ^1^	-	3.0 ± 0.31	-
22	*n*-nonadecane	C_19_H_40_	ALK	1900	1900 ^1^	-	1.5 ± 0.30	-
23	*n*-heneicosane	C_21_H_44_	ALK	2100	2100 ^1^	-	50.5 ± 2.90	-
24	*n*-tricosane	C_23_H_48_	AKL	2300	2300 ^1^	-	6.3 ± 0.80	-
25	*n*-tetracosane	C_24_H_50_	ALK	2400	2400 ^1^	-	13.4 ± 1.82	-
26	*n*-pentacosane	C25H52	ALK	2500	2500 ^1^	-	20.0 ± 0.76	-
	Number of Identified Compounds					6	8	15
	Class of Compounds					VOC-Fs	EOs	VOC-Hyd
	Monoterpene Hydrocarbons (MHs)					-	-	9.9 ± 0.09
	Oxygenated Monoterpenes (OMs)					-	-	4.6 ± 0.19
	Sesquiterpene Hydrocarbons (SHs)					8.3 ± 0.12	-	-
	Oxygenated Sesquiterpenes (OSs)					8.8 ± 0.10	0.7 ± 0.19	-
	Apocarotenoids (ACs)					-	-	0.8 ± 0.04
	Aldehydes (ADHs)					25.7 ± 1.32	3.4 ± 0.44	65.6 ± 1.10
	Alcohols (ALCs)					-	-	6.0 ± 0.30
	Alkanes (ALKs)					33.7 ± 0.32	91.7 ± 1.30	-
	Chromene Compounds (CHRs)					-	3.0 ± 0.31	-
	Esters (ESTs)					7.6 ± 0.14	-	-
	Ketones (KETs)					13.7 ± 0.52	-	1.0 ± 0.14
	Nitrogenous Compounds (NCs)					-	-	9.8 ± 0.79
	Non-terpenes					80.7 ± 0.58	98.1 ± 0.69	82.4 ± 0.59
	Total Identified					97.8 ± 0.24	98.8 ± 0.57	97.7 ± 0.35

LRI ^cal^: Linear Retention Index calculated LRI ^lit^; Linear Retention Index reported in the literature; ^1^: NIST 2014 (National Institute of Standards Technology (www.nist.gov) visited 24 February 2024); * Pherobase.com.

**Table 3 plants-13-01145-t003:** Analysis of spontaneous emissions of fresh flowers (VOC-Fs), essential oils (EOs), and hydrosols (VOC-Hyds) derived from *Calendula officinalis*.

No.	Compounds	Formula	Class	LRI ^cal^	LRI ^lit^	SEs	EOs	VOC-Hyd
						Relative Abundance (%)		
1	ethyl acetate	C_4_H_8_O_2_	EST	743		-	-	1.8 ± 0.02
2	methoxy-phenyl-oxime	C_8_H_9_NO_2_	NC	898	899 *	-	-	5.3 ± 0.36
3	α-thujene	C_10_H_16_	MH	933	931 ^1^	44.8 ± 3.44	-	-
4	β-thujene	C_10_H_16_	MH	976	978 ^1^	1.2 ± 0.15	-	-
5	sabinene	C_10_H_16_	MH	977	976 ^1^	0.2 ± 0.01	-	-
6	β-myrcene	C_10_H_16_	MH	991	990 ^1^	1.0 ± 0.11	-	-
7	*o*-cymene	C_10_H_14_	MH	1022	1020 ^1^	0.4 ± 0.03	-	-
8	*p*-cymene	C_10_H_14_	MH	1028	1026 ^1^	-	-	3.0 ± 0.00
9	eucalyptol	C_10_H_18_O	OM	1032	1033 ^1^	0.2 ± 0.01	-	41.4 ± 0.34
10	limonene	C_10_H_16_	MH	1029	1033 ^1^	0.4 ± 0.04	-	5.6 ± 0.15
11	*cis*-β-ocimene	C_10_H_16_	MH	1038	1039 ^1^	0.3 ± 0.00	-	-
12	γ-terpinene	C_10_H_16_	MH	1058	1053 ^1^	1.2 ± 0.23	-	0.8 ± 0.02
13	fenchone	C_10_H_16_O	OM	1096	1097 ^1^	-	-	1.4 ± 0.02
14	linalool	C_10_H_18_O	OM	1101	1094 ^1^	-	-	12.2 ± 0.33
15	α-thujone	C_10_H_16_O	OM	1103	1102 ^1^	-	-	4.2 ± 0.27
16	nonanal	C_9_H_18_O	ALD	1104	1101 ^1^	0.7 ± 0.06	-	-
17	β-thujone	C_10_H_16_O	OM	1117	1114 ^1^	-	-	1.4 ± 0.14
18	camphor	C_10_H_16_O	OM	1142	1143 ^1^	-	-	6.9 ± 0.05
19	citronellal	C_10_H_18_O	OM	1155	1157 ^1^	-	-	0.9 ± 0.06
20	*cis*-*p*-menthan-3-one	C_10_H_18_O	OM	1166	1164 ^1^	-	-	0.6 ± 0.02
21	thujen-2-one	C_10_H_14_O	OM	1177	1173 ^1^	-	-	0.6 ± 0.03
22	4-terpineol	C_10_H_18_O	OM	1177	1171 ^1^	-	-	6.4 ± 0.06
23	α-terpineol	C_10_H_18_O	OM	1191	1189 ^1^	-	-	0.7 ± 0.02
24	2-propylheptanol	C_10_H_22_O	ALC	1193	1194 ^2^	-	-	1.3 ± 0.08
25	estragole	C_10_H_12_O	pp	1196	1195 ^1^	-	-	1.5 ± 0.06
26	decanal	C_10_H_20_O	ALD	1206	1208 ^1^	1.6 ± 0.02	-	0.5 ± 0.03
27	3,5-dimethyl-2-isobutyl-pyrazine	C_10_H_16_N_2_	NTN	1210	1211 ^1^	0.6 ± 0.04	-	-
28	pulegone	C_10_H_16_O	OM	1240	1237 ^1^	-	-	0.4 ± 0.03
29	methyl carvacrol	C_11_H_16_O	OM	1244	1245 ^1^	-	-	0.8 ± 0.02
30	5-undecen-4-one	C_11_H_20_	KET	1250	1259 ^2^	-	0.2 ± 0.02	-
31	linalyl acetate	C_12_H_20_O_2_	OM	1259	1257 ^1^	-	-	0.3 ± 0.02
32	bornyl acetate	C_12_H_20_O_2_	OM	1285	1284 ^1^	-	-	1.7 ± 0.02
33	α-copaene	C_15_H_24_	SH	1376	1372 ^1^	1.3 ± 0.44	-	-
34	β-caryophyllene	C_15_H_24_	SH	1419	1418 ^1^	0.9 ± 0.03	-	-
35	humulene	C_15_H_24_	SH	1454	1455 ^1^	2.5 ± 0.07	-	-
36	γ-muurolene	C_15_H_24_	SH	1477	1477 ^1^	1.3 ± 0.18	0.2 ± 0.02	-
37	germacrene d	C_15_H_24_	SH	1481	1480 ^1^	4.1 ± 0.51	0.5 ± 0.07	-
38	β-selinene	C_15_H_24_	SH	1486	1486 ^1^	0.3 ± 0.09	-	-
39	*epi*-cubebol	C_15_H_24_O	OS	1493	1494 ^1^	-	1.5 ± 0.21	-
40	α-muurolene	C_15_H_24_	SH	1499	1499 ^1^	2.1 ± 0.07	1.3 ± 0.13	-
41	γ-cadinene	C_15_H_24_	SH	1513	1513 ^1^	11.1 ± 1.14	-	-
42	cubebol	C_15_H_26_O	OS	1515	1516 ^1^	-	1.7 ± 0.16	-
43	δ-cadinene	C_15_H_24_	SH	1524	1524 ^1^	15.3 ± 1.56	15.0 ± 0.73	0.2 ± 0.04
44	α-cadinene	C_15_H_24_	SH	1538	1541 ^1^	0.7 ± 0.09	-	-
45	germacrene d-4-ol	C_15_H_26_O	OS	1575	1578 ^1^	-	1.7 ± 0.17	-
46	1-*epi*-cubenol	C_15_H_26_O	OS	1627	1623 ^1^	-	0.8 ± 0.09	-
47	*tau*-cadinol	C_15_H_26_O	OS	1641	1638 ^1^	1.4 ± 0.21	16.1 ± 1.45	-
48	*tau*-muurolol	C_15_H_26_O	OS	1646	1642 ^1^	-	2.1 ± 0.15	-
49	α-cadinol	C_15_H_26_O	OS	1653	1653 ^1^	0.5 ± 0.05	18.8 ± 1.90	-
50	hexahydrofarnesylacetone	C_18_H_36_O	AC	1844	1847 ^1^	-	0.6 ± 0.09	-
51	nonadecane	C_19_H_40_	ALK	1900	1900 ^1^	1.3 ± 0.27	0.2 ± 0.03	-
52	methyl linolenate	C_19_H_32_O_2_	FA	2098	2101 ^1^	-	0.3 ± 0.07	-
53	heneicosane	C_21_H_44_	ALK	2100	2100 ^1^	-	0.3 ± 0.06	-
54	linolenic acid	C_18_H_30_O_2_	FA	2139	2143 ^1^	-	1.0 ± 0.17	-
55	dodecyl caprylate	C_20_H_40_O_2_	EST	2160	2160 ^1^	0.2 ± 0.05	-	-
56	octadecyl acetate	C_20_H_40_O_2_	EST	2208	2211 ^1^	-	0.3 ± 0.03	-
57	3-methylbutyl hexadecanoate	C_21_H_42_O_2_	EST	2253	2260 ^1^	-	11.0 ± 1.41	-
58	methyl arachidonate	C_21_H_34_O_2_	FA	2255	2274 ^1^	-	0.2 ± 0.05	-
59	tetracosane	C_24_H_50_	ALK	2300	2400 ^1^	-	0.4 ± 0.14	-
60	(*Z,Z,Z*)-8,11,14-eicosatrienoic acid	C_20_H_34_O_2_	FA	2346	2347 ^1^	-	1.1 ± 0.12	-
61	hexyl heptadecanoate	C_23_H_46_O_2_	EST	2464	2464 ^1^	-	8.2 ± 0.43	-
62	pentacosane	C_25_H_52_	ALK	2500	2500 ^1^	-	6.0 ± 0.91	-
	Number of Identified compounds					26	24	24
	Class of Compounds					VOC-Fs	EOs	VOC-Hyd
	Monoterpene Hydrocarbons (MHs)					49.5 ± 2.13	-	9.4 ± 0.17
	Oxygenated Monoterpenes (OMs)					0.2 ± 0.01	-	79.9 ± 0.12
	Sesquiterpene Hydrocarbons (SHs)					39.6 ± 1.84	17.0 ± 0.94	0.2 ± 0.04
	Oxygenated Sesquiterpenes (OSs)					1.9 ± 0.20	42.7 ± 2.25	-
	Apocarotenoids (ACs)					-	-	-
	Phenylpropanoids (PPs)					-	-	1.5 ± 0.06
	Alcohols (ACLs)					-	-	1.3 ± 0.08
	Aldehydes (ALDs)					2.3 ± 0.03	-	0.5 ± 0.03
	Alkanes (ALKs)					1.3 ± 0.27	6.9 ± 0.13	-
	Esters (ESTs)					0.2 ± 0.05	19.5 ± 0.75	1.8 ± 0.02
	Fatty acids (FAs)					-	2.6 ± 0.13	-
	Ketones (KETs)					-	0.2 ± 0.02	-
	Nitrogenous Compounds (NCs)					-	-	5.3 ± 0.36
	Nitrogenous Compunds (NTNs)					0.6 ± 0.04	-	-
	Non-terpenes					4.4 ± 0.33	29.2 ± 2.28	8.9 ± 0.26
	Total Identified					95.7 ± 1.25	89.5 ±1.25	99.9 ± 0.05

LRI ^cal^: Linear Retention Index calculated LRI ^lit^; Linear Retention Index reported in the literature; ^2^: Chemspider; ^1^: NIST 2014 (National Institute of Standards Technology (www.nist.gov) visited 24 February 2024). * Pherobase.com.

**Table 5 plants-13-01145-t005:** Analysis of spontaneous emissions of fresh flowers (VOC-Fs), essential oils (EO), and hydrosols (VOC-Hyd) derived from *Polianthes tuberosa*.

No.	Compounds	Formula	Class	LRI ^cal^	LRI ^lit^	VOC-Fs	EOs	VOC-Hyd
						Relative Abundance (%)		
1	2,4,5-trimethyl oxazole	C_6_H_9_NO	NC	852	850 ^2^	-	-	7.4 ±0.51
2	4-heptanone	C_7_H_14_O	KET	871	872 ^2^	-	-	0.6 ±0.01
3	methoxy-phenyl oxime	C_8_H_9_NO_2_	NC	898	899 *	-	-	15.5 ± 0.91
4	α-pinene	C_10_H_16_	MH	941	937 ^2^	-	-	0.8 ± 0.04
5	benzaldehyde	C_7_H_6_O	ADH	962	969 ^1^	-	-	0.5 ± 0.07
6	β-pinene	C_10_H_16_	MH	982	980 ^1^	-	-	0.8 ± 0.04
7	6-methyl- 5-hepten-2-one	C_8_H_14_O	KET	986	987 *	-	-	4.5 ± 0.06
8	*p*-cymene	C_10_H_14_	MH	1028	1026 ^1^	-	-	8.9 ± 0.05
9	limonene	C_10_H_16_	MH	1029	1033 ^1^	-	-	5.7 ± 0.54
10	eucalyptol	C_10_H_18_O	OM	1032	1033 ^1^	1.6 ± 0.33	-	38.1 ± 0.3
11	γ-terpinene	C_10_H_16_	MH	1058	1053 ^1^	-	-	0.6 ± 0.01
12	1-octanol	C_8_H_18_O	ALC	1071	1074 ^1^	-	-	0.4 ± 0.05
13	methyl benzoate	C_8_H_8_O_2_	EST	1092	1091 ^1^	-	7.3 ± 0.20	-
14	methyl ester benzoic acid (=clorius = niobe oil = methyl benzoate)	C_8_H_8_O_2_	EST	1094	1091 ^1^	57.3 ± 3.20	-	4.2 ± 0.05
15	nonanal	C_9_H_18_O	AD	1104	1101 ^1^	-	-	2.8 ± 0.06
16	menthone	C_10_H_18_O	OM	1154	1155 ^1^	-	-	0.2 ± 0.14
17	α-terpineol	C_10_H_18_O	OM	1191	1198 ^1^	-	-	1.2 ± 0.07
18	methyl salicylate	C_8_H_8_O_3_	EST	1192	1190 ^1^	13.0 ± 2.57	-	-
19	estragol	C_10_H_12_O	OM	1196	1997 ^1^	-	-	0.8 ± 0.02
20	decanal	C_10_H_20_O	ADH	1206	1208 ^1^	-	-	2.4 ± 0.06
21	(*E*)-anethole	C_10_H_12_O_2_	PP	1286	1284 ^1^	-	-	0.2 ± 0.13
22	methyleugenol	C_11_H_14_O_2_	PP	1405	1404 ^1^	0.6 ± 0.02	-	0.8 ± 0.00
23	geranylacetone	C_13_H_22_O	AC	1456	1457 ^1^	-	-	0.2 ± 0.03
24	methyl ether *iso*-eugenyl	C_11_H_14_O_2_	PP	1492	1494 ^1^	5.9 ± 0.21	-	-
25	methyl ether iso-eugenol	C_11_H_14_O_2_	PP	1492	1494 ^1^	-	-	2.2 ± 0.09
26	δ-decalactone	C_10_H_18_O_2_	LAC	1496	1497 ^1^	0.6 ± 0.05	-	-
27	jasminelactone	C_10_H_16_O_2_	LAC	1518		13.8 ± 1.80	-	-
28	precocene ii	C_13_H_16_O_3_	CHR	1658	1656 ^1^	-	4.0 ± 0.00	-
29	xanthorrhizol	C_15_H_22_O	OS	1753	1754 ^1^		2.9 ± 0.10	-
30	benzyl benzoate	C_14_H_12_O_2_	EST	1762	1765 ^1^	5.5 ± 0.56	-	-
31	1-hexadecanol	C16H34O	ALC	1880	1881 ^1^	0.7 ± 0.06	-	-
32	methyl icosanoate (=methyl arachidate)	C_21_H_42_O_2_	EST	2329	2324 ^1^	-	24.4 ± 0.83	-
33	methyl heneicosanoate	C_22_H_44_O_2_	EST	2429	2428 ^1^		58.4 ± 0.65	-
	Number of Identified compounds					9	5	22
	Class of Compounds					VOC-Fs	EOs	VOC-Hyd
	Monoterpene Hydrocarbons (MHs)					-	-	16.8 ± 0.2
	Oxygenated Monoterpenes (OMs)					1.6 ± 0.33	-	40.3 ± 0.50
	Apocarotenoids (ACs)					-	-	0.2 ± 0.03
	Phenylpropanoids (PPs)					6.5 ± 0.10	0,0	3.2 ± 0.30
	Aldehydes (ADHs)					-	-	5.7 ± 0.06
	Alcohols (ALCs)					0.7 ± 0.06	-	0.4 ± 0.05
	Esters (ESTs)					76.5 ± 1.80	90.1 ± 1.20	4.6 ± 0.10
	Ketones (KETs)					-	-	5.1 ± 0.03
	Nitrogenous Compounds (NCs)					-	-	22.9 ± 0.70
	Lactones Compounds (LACs)					14.4 ± 1.70	-	-
	Chromene Compounds (CHRs)					-	4.0 ± 0.00	-
	Non-terpenes					91.6 ± 1.2	94.1 ± 0.78	38.7 ± 0.28
	Total Identified					99.7 ± 1.10	94.1 ± 1.20	99.2 ± 0.30

LRI ^cal^: Linear Retention Index calculated LRI ^lit^; Linear Retention Index reported in the literature; ^2^: Chemspider; ^1^: NIST 2014 (National Institute of Standards Technology (www.nist.gov) visited 24 February 2024); * Pherobase.com.

**Table 6 plants-13-01145-t006:** Analysis of spontaneous emissions of fresh flowers (VOC-Fs), essential oils (EO), and hydrosols (VOC-Hyd) derived from *Tulbaghia cominsii*.

No.	Compounds	Formula	Class	LRI ^cal^	LRI ^lit^	VOC-Fs	EOs	VOC-Hyd
						Relative Abundance (%)		
1	3-hexanol	C_6_H_14_O	ALC	797	801 ^1^	-	-	1.20.21
2	2-hexanol	C_6_H_14_O	ALC	801	800 ^1^	-	-	2.1 ± 0.12
3	methoxy-phenyl-oxime	C_8_H_9_NO_2_	NC	898	899 *	-	-	7.5 ± 0.25
4	α-thujene	C_10_H_16_	MH	933	931 ^1^	-	-	-
5	α-pinene	C_10_H_16_	MH	941	937 ^1^	-	0.5 ± 0.09	0.6 ± 0.03
6	benzaldehyde	C_7_H_6_O	ADH	962	969 ^1^	0.5 ± 0.10	-	0.3 ± 0.07
7	β-pinene	C_10_H_16_	MH	982	980 ^1^	-	0.6 ± 0.04	0.7 ± 0.03
8	6-methyl- 5-hepten-2-one	C_8_H_14_O	KET	986	987 ^1^	-	-	0.8 ± 0.02
9	dimethyl trisulfite	C_2_H_6_S_3_	SC	993	982 ^1^	0.3 ± 0.07	1.8 ± 0.22	-
10	*p*-cymene	C_10_H_14_	MH	1028	1026 ^1^	-	-	8.3 ± 0.07
11	limonene	C_10_H_16_	MH	1029	1033 ^1^	-	-	11.4 ± 0.31
12	2-ethyl 1-hexanol	C_8_H_18_O	ALC	1030	1035 ^1^	1.6 ± 0.34	-	-
13	eucalyptol	C_10_H_18_O	OM	1032	1033 ^1^	-	-	21.4 ± 0.24
14	benzyl alcohol	C_7_H_8_O	ALC	1036	1037 ^1^	1.2 ± 0.23	-	-
15	phenylacetaldehyde (=benzeneacetald.)	C_8_H_8_O	ADH	1045	1043 ^1^	-	0.4 ± 0.02	
16	2-propyl-1-pentanol	C_8_H_18_O	ALC	1052	1053 ^1^	0.3 ± 0.08	-	-
17	γ-terpinene	C_10_H_16_	MH	1058	1053 ^1^	-	-	1.1 ± 0.03
18	1-octanol	C_8_H_18_O	ALC	1071	1074 ^1^	-	-	0.4 ± 0.08
19	fenchone	C_10_H_16_O	OM	1096	1097 ^1^	-	-	0.4 ± 0.02
20	linalool	C_10_H_18_O	OM	1101	1094 ^1^	-	-	0.7 ± 0.03
21	thujone	C_10_H_16_O	OM	1103	1102 ^1^	-	-	1.9 ± 0.06
22	nonanal	C_9_H_18_O	ADH	1104	1101 ^1^	1.3 ± 0.53	0.6	5.8 ± 0.01
23	thioanisole	C_7_H_8_S	SC	1106	1106 ^1^	0.9 ± 0.17	-	-
24	camphor	C_10_H_16_O	OM	1142	1143 ^1^	-	-	0.8 ± 0.04
25	disulfide, methyl (methylthio) methyl	C_3_H_8_S_3_	SC	1143	1147 ^1^	2.2 ± 0.58	25.8 ± 0.95	-
26	benzyl nitrite	C_7_H_7_NO_2_	NC	1144	1143 ^1^	1.1 ± 0.11	-	-
27	menthone	C_10_H_18_O	OM	1154	1154 ^1^	-	-	0.8 ± 0.12
28	*p*-anisyl vinyl ether	C_9_H_10_O	ETR	1156	1154 ^1^	0.6 ± 0.13	-	-
29	benzyl acetate	C_9_H_10_O_2_	EST	1164	1162 ^1^	10.6 ± 0.79	-	-
30	borneol	C_10_H_18_O	OM	1167	1168 ^1^	-	-	0.3 ± 0.02
31	4-terpineol	C_10_H_18_O	OM	1177	1171 ^1^	-	-	0.4 ± 0.00
32	estragol	C_10_H_12_O	OM	1196	1997 ^1^	-	-	1.1 ± 0.05
33	decanal	C_10_H_20_O	ADH	1206	1208 ^1^	2.3 ± 0.23	-	9.6 ± 0.18
34	pulegone	C_10_H_16_O	OM	1240	1237 ^1^	-	0.3 ± 0.01	-
35	phenethyl acetate	C_10_H_12_O_2_	EST	1258	1255 ^1^	7.80.91	1.7 ± 0.07	-
36	bornyl acetate	C_12_H_20_O_2_	OM	1285	1284 ^1^	-	1.7 ± 0.02	-
37	(*E*)-anethole	C_10_H_12_O_2_	PP	1286	1284 ^1^	-	-	0.6 ± 0.08
38	thymol	C_10_H_14_O	OM	1291	1292 ^1^	5.1 ± 0.41	16.3 ± 0.21	19.1 ± 0.20
39	undecanal	C_11_H_22_O	ADH	1307	1309 ^1^	-	-	0.7 ± 0.03
40	4-acetylanisol	C_9_H_10_O_2_	KET	1350	1355 ^1^	1.6 ± 0.35	-	-
41	tetradecane	C_14_H_30_	ALK	1400	1400 ^1^	2.4 ± 0.45	-	-
42	β-caryophyllene	C_15_H_24_	SH	1419	1418 ^1^	-	1.2 ± 0.09	-
43	5,9-undecadien-2-one, 6,10-dimethyl- (*trans*-geranylacetone)	C_13_H_22_O	KET	1456	1453 ^1^	0.3 ± 0.07	-	-
44	2,4,5,7-tetrathiaoctane	C_4_H_10_S_4_	SC	1484	1496 *	2.4 ± 0.59	10.0 ± 0.40	-
45	ethanone, 1-(3,4-dimethoxyphenyl)- (=acetoveratrone)	C_10_H_12_O_3_	KET	1569	1573 ^1^	28.4 ± 0.67	-	-
46	viridiflorol	C_15_H_26_O	OS	1591	1593 ^1^	1.1 ± 0.27	2.7 ± 0.17	-
47	hedione	C_13_H_22_O_3_	EST	1649	1648 ^1^	0.6 ± 0.25	-	-
48	precocene ii	C_13_H_16_O_3_	CHR	1658	1656 ^1^	-	6.3 ± 0.06	-
49	epi-α-bisabolool	C_15_H_26_O	OS	1684	1684 ^1^	-	0.7 ± 0.06	-
50	benzyl benzoate	C_14_H_12_O_2_	EST	1762	1765 ^1^	14.5 ± 1.56	1.0 ± 0.07	-
51	2,3,5,7-tetrathioctane 3,3dioxide	C_4_H_10_O_2_S_4_	SC	1784	1783 ^1^	-	0.5 ± 0.11	-
52	methyl (*E*,*E*)-farnesoate	C_16_h_26_O_2_	OS	1786	1789 ^1^	0.4 ± 0.08	-	-
53	2,4,5,6,8-pentathianonane	C_4_H_10_S_5_	NC	1853	1852 ^1^	0.6 ± 0.30	-	-
54	phenethyl benzoate	C_15_H_14_O_2_	EST	1856	1858 ^2^	2.4 ± 0.43	-	-
55	benzyl salicylate	C_14_H_12_O_3_	EST	1869	1863 ^1^	0.80.08	-	-
56	*n*-heneicosane	C_21_H_44_	ALK	2100	2100 ^1^	-	22.9 ± 0.16	-
57	dodecyl octanoate	C_20_H_40_O_2_	EST	2160	2160 ^1^	0.5 ± 0.08	-	-
58	octadecanoic acid (stearic acid)	C_18_H_36_O_2_	FA	2172	2177 ^1^	0.2 ± 0.08	-	-
59	phenyl ethyl alcohol	C_8_H_10_O	ALC	1114		3.6 ± 0.89	-	-
60	Number of Identified Compounds					29	18	25
	Class of Compounds					VOC-Fs	EOs	VOC-Hyd
	Monoterpene Hydrocarbons (MHs)					-	1.1 ± 0.10	22.1 ± 0.50
	Oxygenated Monoterpenes (OMs)					5.1 ± 0.40	18.3 ± 0.20	46.9 ± 0.80
	Sesquiterpene Hydrocarbons (SHs)					-	1.2 ± 0.09	-
	Oxygenated Sesquiterpenes (OSs)					1.5 ± 0.22	3.4 ± 0.30	-
	Phenylpropanoids (PPs)					-	-	0.6 ± 0.08
	Aldehydes (ADHs)					4.1 ± 0.60	1.0 ± 0.10	16.4 ± 0.30
	Alcohols (ALCs)					6.7 ± 0.70	-	3.7 ± 0.40
	Alkanes (ALKs)					2.4 ± 0.45	22.9 ± 0.16	-
	Chromene Compounds (CHRs)					-	6.3 ± 0.06	-
	Esters (ESTs)					37.2 ± 1.30	2.7 ± 0.10	-
	Ethers (ETRs)					0.6 ± 0.13	-	-
	Fatty Acid (FAs)					0.2 ± 0.08	-	-
	Ketones (KETs)					30.3 ± 0.35	-	0.8 ± 0.02
	Nitrogenous Compounds (NCs)					1.7 ± 0.30	-	7.5 ± 0.25
	Sulfurus Compounds (SCs)					5.8 ± 0.60	38.1 ± 0.43	-
	Non-terpenes					89.0 ± 0.52	71.0 ± 0.17	28.4 ± 0.25
	Total Identified					95.6 ± 0.40	95.0 ± 0.20	98.0 ± 0.30

LRI ^cal^: Linear Retention Index calculated LRI ^lit^; Linear Retention Index reported in the literature; ^2^: Chemspider; ^1^: NIST 2014 (National Institute of Standards Technology (www.nist.gov) visited 24 February 2024); * Pherobase.com.

**Table 7 plants-13-01145-t007:** Antibacterial activity of flower-derived hydrosols.

Plant Species			MIC1	MIC2	MIC3	MIC Mode	MBC1	MBC2	MBC3	MBC Mode
*A. majus*	*Gram-positive*	*Enterococcus faecalis ATCC VAN B V583 E*	>1:2	>1:2	>1:2	**>1:2**	>1:2	>1:2	>1:2	**>1:2**
		*Listeria monocytogenes ATCC 7644*	>1:2	>1:2	>1:2	**>1:2**	>1:2	>1:2	>1:2	**>1:2**
		*Staphylococcus aureus* ATCC 6538	1:2	>1:2	1:2	**1:2**	>1:2	>1:2	>1:2	**>1:2**
	*Gram-negative*	*Escherichia coli ATCC 15325*	>1:2	>1:2	>1:2	**>1:2**	>1:2	>1:2	>1:2	**>1:2**
		*Pseudomonas aeruginosa ATCC 27853*	>1:2	>1:2	>1:2	**>1:2**	>1:2	>1:2	>1:2	**>1:2**
		*Salmonella enterica ser. Typhimurium ATCC 14028*	>1:2	>1:2	>1:2	**>1:2**	>1:2	>1:2	>1:2	**>1:2**
*B cucullata*	*Gram-positive*	*Enterococcus faecalis ATCC VAN B V583 E*	>1:2	>1:2	>1:2	**>1:2**	>1:2	>1:2	>1:2	**>1:2**
		*Listeria monocytogenes ATCC 7644*	>1:2	>1:2	>1:2	**>1:2**	>1:2	>1:2	>1:2	**>1:2**
		*Staphylococcus aureus* ATCC 6538	1:2	>1:2	1:4	**1:2**	>1:2	1:2	>1:2	**>1:2**
	*Gram-negative*	*Escherichia coli ATCC 15325*	>1:2	>1:2	>1:2	**>1:2**	>1:2	>1:2	>1:2	**>1:2**
		*Pseudomonas aeruginosa ATCC 27853*	>1:2	>1:2	>1:2	**>1:2**	**>1:2**	>1:2	>1:2	**>1:2**
		*Salmonella enterica ser. Typhimurium ATCC 14028*	1:2	>1:2	>1:2	**1:2**	>1:2	1:2	>1:2	**>1:2**
*C. officinalis*	*Gram-positive*	*Enterococcus faecalis ATCC VAN B V583 E*	>1:2	>1:2	>1:2	**>1:2**	>1:2	>1:2	>1:2	**>1:2**
		*Listeria monocytogenes ATCC 7644*	>1:2	>1:2	>1:2	**>1:2**	>1:2	>1:2	>1:2	**>1:2**
		*Staphylococcus aureus* ATCC 6538	1:2	>1:2	1:4	**1:2**	>1:2	1:2	>1:2	**>1:2**
	*Gram-negative*	*Escherichia coli ATCC 15325*	>1:2	>1:2	>1:2	**>1:2**	>1:2	>1:2	>1:2	**>1:2**
		*Pseudomonas aeruginosa ATCC 27853*	>1:2	>1:2	>1:2	**>1:2**	**>1:2**	>1:2	>1:2	**>1:2**
		*Salmonella enterica ser. Typhimurium ATCC 14028*	1:2	>1:2	>1:2	**>1:2**	>1:2	1:2	>1:2	**>1:2**
*D*. *hortensis*	*Gram-positive*	*Enterococcus faecalis ATCC VAN B V583 E*	>1:2	>1:2	>1:2	**>1:2**	>1:2	>1:2	>1:2	**>1:2**
		*Listeria monocytogenes ATCC 7644*	>1:2	>1:2	>1:2	**>1:2**	>1:2	>1:2	>1:2	**>1:2**
		*Staphylococcus aureus* ATCC 6538	1:2	>1:2	1:4	**>1:2**	1:2	>1:2	1:2	**>1:2**
	*Gram-negative*	*Escherichia coli ATCC 15325*	>1:2	>1:2	>1:2	**>1:2**	>1:2	>1:2	>1:2	**>1:2**
		*Pseudomonas aeruginosa ATCC 27853*	>1:2	>1:2	>1:2	**>1:2**	>1:2	>1:2	>1:2	**>1:2**
		*Salmonella enterica ser. Typhimurium ATCC 14028*	1:2	>1:2	>1:2	**1:2**	>1:2	1:2	>1:2	**>1:2**
*P. tuberosa*	*Gram- positive*	*Enterococcus faecalis ATCC VAN B V583 E*	>1:2	>1:2	>1:2	**>1:2**	>1:2	>1:2	>1:2	**>1:2**
		*Listeria monocytogenes ATCC 7644*	>1:2	>1:2	>1:2	**>1:2**	>1:2	>1:2	>1:2	**>1:2**
		*Staphylococcus aureus* ATCC 6538	1:2	>1:2	1:4	**1:2**	>1:2	1:2	>1:2	**>1:2**
	*Gram- negative*	*Escherichia coli ATCC 15325*	>1:2	>1:2	>1:2	**>1:2**	>1:2	>1:2	>1:2	**>1:2**
		*Pseudomonas aeruginosa ATCC 27853*	>1:2	>1:2	>1:2	**>1:2**	>1:2	>1:2	>1:2	**>1:2**
		*Salmonella enterica ser. Typhimurium ATCC 14028*	1:2	>1:2	>1:2	**1:2**	>1:2	1:2	>1:2	**>1:2**
*T. cominsii*	*Gram- positive*	*Enterococcus faecalis ATCC VAN B V583 E*	>1:2	>1:2	>1:2	**>1:2**	>1:2	>1:2	>1:2	**>1:2**
		*Listeria monocytogenes ATCC 7644*	>1:2	>1:2	>1:2	**>1:2**	>1:2	>1:2	>1:2	**>1:2**
		*Staphylococcus aureus* ATCC 6538	>1:2	>1:2	>1:2	**>1:2**	>1:2	1:2	>1:2	**>1:2**
	*Gram- negative*	*Escherichia coli ATCC 15325*	>1:2	>1:2	1:2	**1:2**	>1:2	1:2	>1:2	**>1:2**
		*Pseudomonas aeruginosa ATCC 27853*	>1:2	>1:2	>1:2	**>1:2**	>1:2	>1:2	>1:2	**>1:2**
		*Salmonella enterica ser. Typhimurium ATCC 14028*	1:2	>1:2	>1:2	**1:2**	>1:2	1:2	>1:2	**>1:2**

Values and mode (in bold) of Minimum Inhibitory Concentration (MIC) and Minimum Bactericidal Concentration (MBC) of each hydrosol against the six ATCC bacterial strains used in the test.

## Data Availability

Data are contained within the article.

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
