# Peer review of "Exploring the Volatile Composition and Antibacterial Activity of Edible Flower Hydrosols with Insights into Their Spontaneous Emissions and Essential Oil Chemistry"

_plants, 2024, doi:10.3390/plants13081145_

Round 1
Reviewer 1 Report
Comments and Suggestions for Authors
Overall, the paper needs a lot of work. The English is not at a publication level. There are several spelling mistakes or incorrect phrasings throughout the entire paper. I tried to highlight as many as I could but there were too many. Please proofread the paper several times again prior to submission. Additionally, be consistent with capitalization. Sometimes compounds are capitalized and sometimes they are not.
Secondly, the data analysis section is very questionable. How did you identify the compounds? There was no mention of a standard used so it can’t be positive identification. Were the peaks selected based on their retention index alone or in addition to the mass spectra used? Furthermore, the section containing the PCA is weak. It is clear the authors do not quite understand how PCA works. Additionally, there is no mention in the materials and methods of how this PCA plot was produced. It is critical to normalize data and to scale it properly prior to data analysis.
For the experiment work, was an internal standard used anywhere during the experiment? There was no mention. There is also no mention on how the EO were extracted, this needs to be included.
Please see below for corrections/comments to specific regions of the paper.
Line 78-79: remove dash from composition
Line 122: already abbreviated EO, no need to write it out and abbreviate it again
Line 183: change aldehyde to aldehydes
Line 265: remove is, replace differ with differs
Line 267: who is Kutty? Improper citation
Line 274: be consistent with using EO or writing essential oil in the manuscript
Line 278: oxygenated monoterpenes have already been abbreviated, be consistent with abbreviations
Line 279: nitrogenous compounds have already been abbreviated, be consistent with abbreviations
Line 280: T in 2,4,5-trimethyl oxazole does not need to be capitalized
Lines 279-281: This sentence does not make sense. Please rephrase it.
Line 281-283: Needs to be rephrased, the beginning of it does not make sense
Line 289: add the word “of” before T. cominsii’s
Line 291: change “was” to “were”
Line 296: add “the” between “as” and “main”
Line 298: change “alkane” to “alkanes”
Line 301: change “monoterpenes” to “monoterpene”
Line 303-304 needs to be rephrased, it does not make any sense
Line 305: change “of” to “for”
Line 362: change “provide” to “provides”
Line 363: change to “According to the PCA plot”
Line 370: change “Exanimating” to “Examining”
Line 374: reference missing for p-cymene
Line 382: remove “Interesting”
Section 2.3
-
How were the EO extracted? It only mentions fresh edible flower or hydrosols
-
Make it clear that the VOCs (line 112) are coming from fresh flowers as VOCs are also coming from hydrosols
Section 2.4
-
Change analyses of VOCs to fresh flowers as again, VOCs can come from essential oils and hydrosols (line 122)
Section 3.1.3
-
Emphasize that the VOCs are coming from the flowers
-
É‘-thujene doesn't need to be capitalized
-
Why is a comparison to C. arvensis relevant? They are different species so it isn’t contradicting to have different results than it
Section 3.1.5.
-
Make it clear where these VOCs are coming from (line 289)
-
Line 303-304 needs to be rephrased, it does not make any sense
Section 3.3
-
When referring to the axes for a PCA plot, refer to them as principle component (PC) 1 or 2 instead of first and second axis
-
Change dissimilarity to variability
-
What cluster analysis are you referring to? (Line 366-367)
-
Bold to say they are clustering due to the high amounts of eucalyptol. How do you know it isn’t all the other compounds that are clustering in that quadrant?
-
How was this data normalized and scaled prior performing a PCA? There needs to be a section in the materials and methods discussing how the data was normalized, how it was scaled, what software you used, etc.
Table 7: Change positif to positive and negatif to negative
Comments on the Quality of English Language
Overall, the paper needs a lot of work. The English is not at a publication level. There are several spelling mistakes or incorrect phrasings throughout the entire paper. I tried to highlight as many as I could but there were too many. Please proofread the paper several times again prior to submission. Additionally, be consistent with capitalization. Sometimes compounds are capitalized and sometimes they are not.
Author Response
Overall, the paper needs a lot of work. The English is not at a publication level. There are several spelling mistakes or incorrect phrasings throughout the entire paper. I tried to highlight as many as I could but there were too many. Please proofread the paper several times again prior to submission. Additionally, be consistent with capitalization. Sometimes compounds are capitalized and sometimes they are not.
Answer: We thank the reviewer for their feedback. Regarding the English language, the text was reviewed by a native English speaker.
Secondly, the data analysis section is very questionable. How did you identify the compounds? There was no mention of a standard used so it can’t be positive identification. Were the peaks selected based on their retention index alone or in addition to the mass spectra used? Furthermore, the section containing the PCA is weak. It is clear the authors do not quite understand how PCA works. Additionally, there is no mention in the materials and methods of how this PCA plot was produced. It is critical to normalize data and to scale it properly prior to data analysis.
Thank you for your feedback. We have revised the data analysis section accordingly.
To ensure compound identification, we compared retention times with authentic samples and calculated Kovats Index (KI) values. Additionally, we used computer matching against mass spectra libraries and MS literature data.
Regarding PCA, we've improved clarity and added details on plot generation in the materials and methods section.
We believe these revisions strengthen our analysis. If you have further questions, feel free to ask. For the experiment work, was an internal standard used anywhere during the experiment? There was no mention. There is also no mention on how the EO were extracted, this needs to be included.
Answer: Thank you for your inquiry. The EO extraction method followed the procedures outlined in section 2.2 of our manuscript. Initially, 20 grams of fresh flowers underwent hydrodistillation for 2 hours using a micro-Clevenger-type apparatus. To address potential low EO yield, 1 mL of HPLC-grade n-hexane was added to the separator funnel. After extraction, the EO dissolved in hexane and hydrosol were collected. These two phases were separated based on density difference, with the hexane layer containing the EO being retrieved using a syringe for GC-MS analysis. The hydrosol was stored in a freezer until further analysis.
Please see below for corrections/comments to specific regions of the paper.
Line 78-79: remove dash from composition
Answer: Done
Line 122: already abbreviated EO, no need to write it out and abbreviate it again
Answer: Corrections have been made to ensure consistency in abbreviation usage throughout the manuscript.
Line 183: change aldehyde to aldehydes
Answer: Done
Line 265: remove is, replace differ with differs
Answer: Done
Line 267: who is Kutty? Improper citation
Answer: The citation has been corrected to provide proper credit.
Line 274: be consistent with using EO or writing essential oil in the manuscript
Answer: We have ensured consistency in the usage of "EO" (essential oil) throughout the manuscript.
Line 278: oxygenated monoterpenes have already been abbreviated, be consistent with abbreviations
Answer: We have ensured consistency in the usage of "OM" (oxygenated monoterpenes) throughout the manuscript.
Line 279: nitrogenous compounds have already been abbreviated, be consistent with abbreviations
Answer: We have ensured consistency in the usage of "NC" (nitrogenous compounds) throughout the manuscript.
Line 280: T in 2,4,5-trimethyl oxazole does not need to be capitalized
Answer: We have corrected the capitalization of "T" in "2,4,5-trimethyl oxazole"
Lines 279-281: This sentence does not make sense. Please rephrase it.
Answer: The sentence was rephrased accordingly
Line 281-283: Needs to be rephrased, the beginning of it does not make sense
Answer: The sentence was rephrased accordingly
Line 289: add the word “of” before T. cominsii’s
Answer: Done
Line 291: change “was” to “were”
Answer: Done
Line 296: add “the” between “as” and “main”
Answer: Done
Line 298: change “alkane” to “alkanes”
Answer: Done
Line 301: change “monoterpenes” to “monoterpene”
Answer: Done
Line 303-304 needs to be rephrased, it does not make any sense
Answer: The sentence was rephrased accordingly
Line 305: change “of” to “for”
Answer: Done
Line 362: change “provide” to “provides”
Answer: Done
Line 363: change to “According to the PCA plot”
Answer: Done
Line 370: change “Exanimating” to “Examining”
Answer: Done
Line 374: reference missing for p-cymene
Answer: Done
Line 382: remove “Interesting”
Answer: the sentence was rephrased.
Section 2.3
- How were the EO extracted? It only mentions fresh edible flower or hydrosols
Answer: Thank you for your inquiry. The EO extraction method followed the procedures outlined in section 2.2 of our manuscript. Initially, 20 grams of fresh flowers underwent hydrodistillation for 2 hours using a micro-Clevenger-type apparatus. To address potential low EO yield, 1 mL of HPLC-grade n-hexane was added to the separator funnel. After extraction, the EO dissolved in hexane and hydrosol were collected. These two phases were separated based on density difference, with the hexane layer containing the EO being retrieved using a syringe and directly analyzed by GC-MS analysis. The hydrosol was stored in a freezer until further analysis.
- Make it clear that the VOCs (line 112) are coming from fresh flowers as VOCs are also coming from hydrosols
Answer: Thank you for your query. Indeed, in line 112, when we refer to VOCs, we are specifically discussing those derived from fresh flowers. To avoid any confusion, we have used the abbreviation "VOC-F" to denote VOCs in fresh flowers and "VOC-Hydr." to indicate VOCs in hydrosols throughout the manuscript.
Section 2.4
- Change analyses of VOCs to fresh flowers as again, VOCs can come from essential oils and hydrosols (line 122)
Answer: Thank you for your query. Indeed, in line 112, when we refer to VOCs, we are specifically discussing those derived from fresh flowers. To avoid any confusion, we have used the abbreviation "VOC-F" to denote VOCs in fresh flowers and "VOC-Hydr." to indicate VOCs in hydrosols throughout the manuscript.
Section 3.1.3
- Emphasize that the VOCs are coming from the flowers
Answer: Thank you for your query. Indeed, in line 112, when we refer to VOCs, we are specifically discussing those derived from fresh flowers. To avoid any confusion, we have used the abbreviation "VOC-F" to denote VOCs in fresh flowers and "VOC-Hydr." to indicate VOCs in hydrosols throughout the manuscript.
- É‘-thujene doesn't need to be capitalized
Answer: We have corrected the capitalization of "T" in " É‘-thujene "
- Why is a comparison to C. arvensis relevant? They are different species so it isn’t contradicting to have different results than it
Answer: We appreciate your observation. Our intention was to compare our findings on hydrolates in our species with those documented in the literature. The only prior research we found regarding hydrolats of this genus pertains to C. arvensis. Accordingly, we have revised the sentence to clarify this point.
Section 3.1.5.
- Make it clear where these VOCs are coming from (line 289)
Answer: Done
- Line 303-304 needs to be rephrased, it does not make any sense
Answer: The sentence was rephrased accordingly.
Section 3.3
- When referring to the axes for a PCA plot, refer to them as principle component (PC) 1 or 2 instead of first and second axis
Answer: Done
- Change dissimilarity to variability
Answer: Done
- What cluster analysis are you referring to? (Line 366-367)
Answer: We have included the missing figure in the manuscript. Thank you for bringing this to our attention.
- Bold to say they are clustering due to the high amounts of eucalyptol. How do you know it isn’t all the other compounds that are clustering in that quadrant?
Answer: We've updated the sentence to reflect that eucalyptol is one of the major compounds contributing to the observed clustering.
- How was this data normalized and scaled prior performing a PCA? There needs to be a section in the materials and methods discussing how the data was normalized, how it was scaled, what software you used, etc.
Answer: Thank you for your inquiry. Detailed information on how the data was normalized, scaled, and the software used for PCA is provided in Section 2.5 of the manuscript.
Table 7: Change positif to positive and negatif to negative
Answer: Done

Reviewer 2 Report
Comments and Suggestions for Authors
The authors have characterized the chemical composition of 6 edible flowers. I think the chemical identification is the best section of this paper, but the bioassay and statistic sections need more work and clarity. The writing style also seems grandiose for what this paper reports.
They have done a good job of identifying compounds with KI values and I assume library match scores, although these scores have not been reported. More detail on the quantification of compounds would be nice. I am assuming that this is based on total ion count in the peak and not a single quantitative ion.
However, there are a number of other issues with this report. The paper has so many spelling and grammatical errors that I did not note them all. The authors often suggest that something is noteworthy, interesting or important without giving any context as to why. Extensive revision of the language needs to be made.
I think the bioassay section also needs more detail. Is there any idea of the actual concentration of the hydrosol? I understand it is a 1:2 dilution but was the original 2mg of the mixture or 2ng? With relative concentrations can you determine this? In the discussion of the biological activity, there is never a mention of the amounts needed for the previously reported activity and if these are anywhere relevant to the amounts found in your samples.
I don’t think a PCA analysis is an advanced analytical tool. The figures are poorly presented as they are and should be improved prior to publication.
Comments on the Quality of English LanguageSpelling issues I found before I stopped writing them all down.
Line 60: I don’t see why this needs to be presented as a list and not just sentences.
Line 68: debris is one word
Line 101: Pharmacy
Author Response
The authors have characterized the chemical composition of 6 edible flowers. I think the chemical identification is the best section of this paper, but the bioassay and statistic sections need more work and clarity. The writing style also seems grandiose for what this paper reports.
They have done a good job of identifying compounds with KI values and I assume library match scores, although these scores have not been reported. More detail on the quantification of compounds would be nice. I am assuming that this is based on total ion count in the peak and not a single quantitative ion.
Answer: We appreciate the reviewer's feedback on our paper. Regarding the quantification of compounds, we would like to clarify that we did not perform quantification in our study. Instead, we provided the relative abundance of the peaks, as indicated in the tables 1-6.
However, there are a number of other issues with this report. The paper has so many spelling and grammatical errors that I did not note them all. The authors often suggest that something is noteworthy, interesting or important without giving any context as to why. Extensive revision of the language needs to be made.
Answer: We thank the reviewer for their feedback. Regarding the English language, the text was reviewed by a native English speaker.
I think the bioassay section also needs more detail. Is there any idea of the actual concentration of the hydrosol? I understand it is a 1:2 dilution but was the original 2mg of the mixture or 2ng? With relative concentrations can you determine this?
Answer: Hydrosol originates as condensation water repeatedly distilled by the process of steam distillation in a continuous current and not as a dry extract to be diluted. Therefore, the starting matrix is liquid and the 1:2 dilution refers to the starting hydrosol diluted in culture broth to allow the bacteria to have sufficient growth medium while in the presence of the highest concentration of test substance.
In the discussion of the biological activity, there is never a mention of the amounts needed for the previously reported activity and if these are anywhere relevant to the amounts found in your samples.
Answer: Thank you for your feedback. We omitted the amounts needed from previous studies as they often utilize essential oils or pure compounds, which differ from our hydrosol samples. Including this information may not offer meaningful insights into our study's findings.
I don’t think a PCA analysis is an advanced analytical tool. The figures are poorly presented as they are and should be improved prior to publication.
Answer: We respectfully disagree with the reviewer's assessment that PCA analysis is not an advanced analytical tool. In our study, PCA and HCA analysis proved to be invaluable in providing insights into the distinctive compounds present in each flower. These techniques allowed us to discern relationships within our dataset, which consists of more than 300 matrices. Using such techniques enabled us to condense complex data into comprehensible visual representations.

Reviewer 3 Report
Comments and Suggestions for Authors
This study analyzed hydrosols from six edible flowers. The authors investigated their chemical composition and antibacterial properties, alongside volatile organic compounds (VOCs), and essential oils (EOs). In general, the experiments were well-performed and the manuscript was well-written. It may be published pending some minor revisions.
1) Please do not use currency symbols (e.g. £ and $) in the tables. By the way, what is the full name of "NIST 2014"?
2) The legend for Table 7 is missing. What are the ratios? What are MIC and MBC respectively? Why are some ratios in bold?
3) The letters in Figure 1 are too small to be seen clearly.
1. What is the main question addressed by the research?
The main volatile organic compounds and essential oil compounds in six edible flowers and what are their antibacterial properties?
2. What parts do you consider original or relevant for the field? What specific gap in the field does the paper address?
The study scrutinized the primary phytochemicals inherent in these floral extracts, marking a pivotal stride in revealing the transformative potential encapsulated within these often-neglected by-products.
3. What does it add to the subject area compared with other published material?
The studied hydrosols offer a rich array of chemical compositions with promising applications across multiple industries, including skincare, fragrance, and therapeutics.
4. What specific improvements should the authors consider regarding the methodology? What further controls should be considered?
The methods are well described. No further control should be considered.
5. Please describe how the conclusions are or are not consistent with the evidence and arguments presented. Please also indicate if all main questions posed were addressed and by which specific experiments.
The conclusions are consistent with the evidences. The main questions have been answered correctly.
6. Are the references appropriate?
The references are appropriate.
7. Please include any additional comments on the tables and figures and quality of the data.
See the comments above.
Author Response
This study analyzed hydrosols from six edible flowers. The authors investigated their chemical composition and antibacterial properties, alongside volatile organic compounds (VOCs), and essential oils (EOs). In general, the experiments were well-performed and the manuscript was well-written. It may be published pending some minor revisions.
- Please do not use currency symbols (e.g. £ and $) in the tables. By the way, what is the full name of "NIST 2014"?
Answer: Thank you for your comment. We have revised the tables accordingly and updated the symbols as suggested by the reviewer. Additionally, we have included the full name of the NIST in the footnotes of the table.
The legend for Table 7 is missing. What are the ratios? What are MIC and MBC respectively? Why are some ratios in bold?
Answer Thank you for your feedback. We have now included the legend for Table 7, which we simply forgot to insert. The legend provides clarification on the MIC and MBC values, ​ and we have highlighted the mode values ​​for both in bold.
3) The letters in Figure 1 are too small to be seen clearly.
Answer: we adjusted the figure one
- What is the main question addressed by the research?
The main volatile organic compounds and essential oil compounds in six edible flowers and what are their antibacterial properties?
- What parts do you consider original or relevant for the field? What specific gap in the field does the paper address?
The study scrutinized the primary phytochemicals inherent in these floral extracts, marking a pivotal stride in revealing the transformative potential encapsulated within these often-neglected by-products.
- What does it add to the subject area compared with other published material?
The studied hydrosols offer a rich array of chemical compositions with promising applications across multiple industries, including skincare, fragrance, and therapeutics.
- What specific improvements should the authors consider regarding the methodology? What further controls should be considered?
The methods are well described. No further control should be considered.
- Please describe how the conclusions are or are not consistent with the evidence and arguments presented. Please also indicate if all main questions posed were addressed and by which specific experiments.
The conclusions are consistent with the evidences. The main questions have been answered correctly.
- Are the references appropriate?
The references are appropriate.
- Please include any additional comments on the tables and figures and quality of the data.

Round 2
Reviewer 1 Report
Comments and Suggestions for Authors
The manuscript have been improved substantially there are a couple of things that have to be address.
1. Section 2.4. What does authentic samples mean for the identification of the compounds?
2. Section 2.5 Authors should rephrase and improve PCA method explanation: PCA method was performed to 6 samples and 58 variables. PCA was plot using the two principal components with the highest variance explained. Mean-center was uses to preprocess the data before PCA.
3. Section 2.2 Authors have not indicated the use of an internal standard.
Comments on the Quality of English LanguageEnglish was improved
Author Response
Answer to referee 1
The manuscript have been improved substantially there are a couple of things that have to be address.
- Section 2.4. What does authentic samples mean for the identification of the compounds?
Answer: For authentic samples we intended pure compounds or samples with a known chemical composition.
- Section 2.5 Authors should rephrase and improve PCA method explanation: PCA method was performed to 6 samples and 58 variables. PCA was plot using the two principal components with the highest variance explained. Mean-center was uses to preprocess the data before PCA.
Answer: We would like to thank the referee for the suggestion. We have rephrased the PCA method explanation.
- Section 2.2 Authors have not indicated the use of an internal standard.
Answer: For the chemical analysis no internal standards were used since we assessed the relative amount of each component, without performing the quantitative absolute analysis.

Reviewer 2 Report
Comments and Suggestions for Authors
The methods section is much improved.
There are some issues with consistency in the writing, sometimes the compounds are written out with a compound number from the table, and other times not. This should be fixed.
The discussion after the PCA section doesn’t seem to come to a unified theme and is simply a list of uses of individual compounds. Apart from presence, it is unclear how the quantity in these uses corresponds to the amount in the hydrosols. It seems that hydrosols could be used as a starting material for further purifications with the data presented in this article.
Line 235: Why is the abbreviation for monoterpene hydrocarbons mentioned here, when the subject was introduced earlier? The addition of hydrocarbon does not seem consistent, in line 236 sesquiterpenes do not have the hydrocarbon addition, but later the abbreviation SH is introduced. My understanding that the default monoterpene is a hydrocarbon, unless a modification is mentioned, such as oxygenated monoterpene. Therefore, monoterpene hydrocarbon is redundant.
Line 366: What did this work find in their study?
Comments on the Quality of English LanguageThere are still a few awkward sentences.
Suggested edits:
Line 124: Helium served as…
Line 172: Is OM introduced earlier in the manuscript, if not needs to be written here.
Line 177: This sentence is hard to read as compounds aren’t mentioned but are in other sections. Also, the wording is difficult.
Author Response
The methods section is much improved.
Answer: We really would like to thanks the referee for the observation.
There are some issues with consistency in the writing, sometimes the compounds are written out with a compound number from the table, and other times not. This should be fixed.
Answer: We would like to thank the referee for noticing this issue. We have added the numbers where missing.
The discussion after the PCA section doesn’t seem to come to a unified theme and is simply a list of uses of individual compounds. Apart from presence, it is unclear how the quantity in these uses corresponds to the amount in the hydrosols. It seems that hydrosols could be used as a starting material for further purifications with the data presented in this article.
Answer: Thank you for your insightful comments and for highlighting this important aspect of our study. While we did not perform quantification of the various compounds present in the hydrosols, it's worth noting that biological activity is often attributed to compounds present in higher relative percentages. However, it's also recognized that compounds present in lower abundance can contribute to overall activity through synergistic or antagonistic interactions with those in higher percentages. In our manuscript, we have focused on identifying and discussing the compounds present in the active hydrosols, drawing upon previous literature to contextualize their activity. We believe this approach provides valuable insights into the biological properties of the hydrosols despite the absence of quantification data.
Line 235: Why is the abbreviation for monoterpene hydrocarbons mentioned here, when the subject was introduced earlier?
Answer: We would like to thank the referee; we have moved the abbreviation after the first mentioned “monoterpene hydrocarbons”.
The addition of hydrocarbon does not seem consistent, in line 236 sesquiterpenes do not have the hydrocarbon addition, but later the abbreviation SH is introduced.
Answer: We have specified that we referred to sesquiterpene hydrocarbons.
My understanding that the default monoterpene is a hydrocarbon, unless a modification is mentioned, such as oxygenated monoterpene. Therefore, monoterpene hydrocarbon is redundant.
Answer: We appreciate the referee's observation. However, although redundant, it is necessary to specify whether the compound belongs to the hydrocarbons or oxygenated form of the terpene classes.
Line 366: What did this work find in their study?
Answer: Thank you for your feedback. We have incorporated the requested information into the manuscript.
Comments on the Quality of English Language
There are still a few awkward sentences.
Suggested edits:
Line 124: Helium served as…
Answer: We have modified the sentence.
Line 172: Is OM introduced earlier in the manuscript, if not needs to be written here.
Answer: Yes, it was introduced at line 71.
Line 177: This sentence is hard to read as compounds aren’t mentioned but are in other sections. Also, the wording is difficult.
Answer: We would like to thank the referee for the observation. We have modified the sentence.
